# Survival Analysis via Density Estimation

## Abstract

This paper introduces an algorithm that reinterprets survival analysis through the lens of density estimation, addressing the challenge of censored inputs inherent to survival data. Recognizing that many survival analysis methodologies are extensions of foundational density estimation models, our approach leverages this intrinsic relationship. By conceptualizing survival analysis as a form of density estimation, our algorithm postprocesses the density estimation outputs to derive survival functions. This framework allows for the application of any density estimation model to effectively estimate survival functions, thereby broadening the toolkit available for survival analysis and enhancing the flexibility and applicability of existing density estimation techniques in this domain. The proposed algorithm not only bridges the methodological gap between density estimation and survival analysis but also offers a versatile and robust approach for handling censored survival data.

## 1 Introduction

Multiclass classification is one of the fundamental tasks in machine learning. The objective of this task is to predict the target label $y \in \mathcal{Y}$ for a feature vector $x \in \mathcal{X}$ given a finite set of samples $(x, y) \overset{\text{i.i.d.}}{\sim} (X, Y)$, where $\mathcal{X}$ and $\mathcal{Y}$ denote the space of feature vectors and the finite set of target labels, respectively, and $X$ and $Y$ are random variables corresponding to $\mathcal{X}$ and $\mathcal{Y}$, respectively. *Density estimation*, or more precisely conditional density estimation, is a variant of multiclass classification where the task is to estimate the probability $\Pr(Y = y | x)$ for all $y \in \mathcal{Y}$ given $x \in \mathcal{X}$. Owing to the numerous applications of density estimation, most multiclass classification libraries are equipped to solve this problem. Examples include the random forest models provided in the `sklearn` package, the gradient boosting models in the `lightgbm` package, and modern neural network models such as ImageNet (Krizhevsky et al., 2012).

Survival analysis, alternatively referred to as time-to-event analysis, is a subfield of statistical studies with extensive applications across various domains such as healthcare, finance, and social sciences (see, e.g., (Wang et al., 2019; Wiegrebe et al., 2024) for survey papers on survival analysis). Survival analysis with $K$ competing risks on discrete times can be formulated as a variant of density estimation. The task can be represented as estimating the probability $\Pr(T_k = t | x)$ for all $k \in [K] = \{1, 2, \ldots, K\}$, $t \in \mathcal{T}$, and $x \in \mathcal{X}$, given a finite set of samples $(x, t, \delta) \overset{\text{i.i.d.}}{\sim} (X, T, \Delta)$. Here, $\mathcal{T} = \{1, 2, \ldots, |\mathcal{T}|\}$ represents the set of discrete times of size $|\mathcal{T}|$, each $T_k$ for $k \in [K]$ is a random variable over the support $\mathcal{T}$, $T = \min_k\{T_1, T_2, \ldots, T_K\}$, and $\Delta = \arg\min_k\{T_1, T_2, \ldots, T_K\}$. The task of survival analysis, namely estimating an individual survival function, corresponds to estimating $\Pr(T_k = t | x)$.

Due to the similarity between density estimation and survival analysis, many methodologies developed for density estimation have been extended to survival analysis, particularly for scenarios with $K = 2$ under the conditional independence assumption (i.e., $T_1 \perp\!\!\!\perp T_2 | X$). For example, the random forests model for density estimation has been extended to random survival forests (Ishwaran et al., 2008) for survival analysis, and modern neural network models for density estimation have been extended to the DeepHit model (Lee et al., 2018) for survival analysis. Regarding loss functions and evaluation metrics, strictly proper scoring rules (Gneiting & Raftery, 2007) for density estimation have been adapted for survival analysis in (Rindt et al., 2022; Yanagisawa, 2023). For calibration metrics, the expected calibration error for density estimation has been extended to D-calibration in (Haider et al., 2020) for survival analysis.

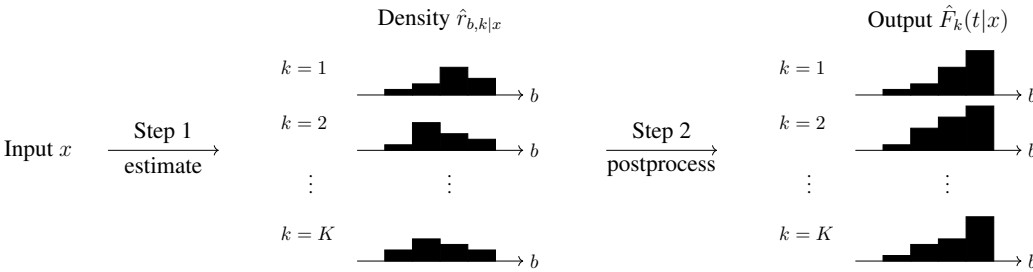

Figure 1: Two-step algorithm for survival analysis with $K$ competing risks: it first estimates $\hat{r}_{b,k|x}$ with an density estimation model and then postprocesses it to obtain the output $\hat{F}_k(t|x)$.

Though numerous extensions of density estimation methodologies for survival analysis exist, these extensions face several limitations. First, these adaptations are tailored to specific methodologies on a case-by-case basis. Hence, when a novel density estimation method arises, a new customized extension must be developed specifically for it. Second, most survival analysis models rely on the conditional independence assumption (or even stronger assumptions such as the proportional hazard assumption (Cox, 1972)), which might not hold in various real-world applications. This underscores the need for survival models that operate under weaker assumptions. Third, Tsiatis (1975) demonstrates that the survival function cannot be identified without assumptions regarding the dependencies between the random variables $T_1, T_2, \ldots, T_k$ (see Sec. E).

In this paper, we address these limitations and explore the following research questions:

**Q1: Can we construct a model-agnostic extension from density estimation to survival analysis?** Yes, we introduce a two-step algorithm for survival analysis that can be integrated with any density estimation model. The first step of this algorithm employs a density estimation model to estimate the joint distribution of the dependent variables. The second step post-processes the results to derive the survival function, as illustrated in Fig. 1. Furthermore, we demonstrate that if the density estimation can achieve an arbitrarily small error $\epsilon$ (as the number of data points increases), our algorithm can estimate survival functions with a small error for $K = 2$ under several plausible assumptions (see Sec. 4).

**Q2: Can we accommodate an assumption weaker than the conditional independence assumption?** Yes, our two-step algorithm can handle dependent variables if prior knowledge of the dependency in the form of a copula is available (see Sec. 2 for an explanation of copulas). Numerous copula-based models for survival analysis manage dependent variables (e.g., (Emura & Chen, 2018)), but these models often incorporate additional assumptions beyond the copula. For instance, the copula-based model proposed in (Gharari et al., 2023) leverages the proportional hazard assumption and is restricted to $K = 2$. Our two-step algorithm, however, relies solely on the copula information to manage dependent variables.

**Q3: Can we estimate the upper and lower bounds of the survival function without prior knowledge of the dependency between random variables?** Yes, a by-product of our two-step algorithm is the ability to estimate the upper and lower bounds of the survival function, accounting for the uncertainty stemming from the lack of knowledge about the copula. This ensures that the two-step algorithm's output for any given copula falls within the estimated upper and lower bounds.

It is important to note that in the context of average survival function estimation (i.e., not the individual survival function), questions Q1–Q3 have already been affirmatively addressed. The Kaplan-Meier estimator (1958), a popular method for estimating the average survival function under the conditional independence assumption, has been extended as the copula-graphic estimator (Zheng & Klein, 1995; Carrière, 1995), which operates under the same assumption as ours that a copula is provided as prior knowledge. The algorithms used in this estimator resemble those in our two-step algorithm. Additionally, our upper and lower bound estimates of the individual survival functions are akin to the average survival function bounds described in (Peterson, 1976).

Lastly, concerning loss functions and evaluation metrics, we introduce a strictly proper scoring rule, Copula-NLL, for copula-based survival analysis. While several proper scoring rules exist for survival analysis with $K = 2$ (Rindt et al., 2022; Yanagisawa, 2023), they are valid only under the conditional independence assumption. Our Copula-NLL is applicable for any $K \geq 2$ and any copula.

## 2 PRELIMINARIES

In this paper, we consider a dataset for survival analysis with $K$ competing risks, represented as $\mathcal{D} = \{(x^{(i)}, t^{(i)}, \delta^{(i)})\}_{i=1}^N$ of size $N$. Here, each $x^{(i)} \in \mathcal{X}$ denotes a feature vector, $t^{(i)} \in \mathcal{T}$ is the observed time, and $\delta^{(i)} \in [K] = \{1, 2, \ldots, K\}$ is the index of the observed risk. It is important to note that in survival analysis, the individual realization times $t_1^{(i)}, t_2^{(i)}, \ldots, t_K^{(i)}$ sampled from the random variables $T_1, T_2, \ldots, T_K$ are unobservable. Instead, we can observe only their minimum value $t^{(i)}$ and the index of the observed risk $\delta^{(i)}$. In this study, we assume that the time horizon is discretized using the boundaries $\{\zeta_b\}_{b=0}^B$ such that $0 = \zeta 0 < \zeta_1 < \cdots < \zeta_B$, where $\zeta_B$ is a sufficiently large number, and we assume that each observed time $t^{(i)}$ satisfies $0 \leq t^{(i)} < \zeta_B$. These assumptions are commonly adopted in numerous survival models (e.g., (Lee et al., 2018; Yanagisawa, 2023; Hickey et al., 2024)). For simplicity, the notation generally excludes $i$, and an observation $(x^{(i)}, t^{(i)}, \delta^{(i)})$ is typically denoted as $(x, t, \delta)$.

The primary task of survival analysis in this study is formulated to estimate the marginal distribution $F_k(\zeta_b|x)$ for each $k \in [K]$, $\zeta_b \in \{\zeta_1, \zeta_2, \ldots, \zeta_{B-1}\}$, and $x \in \mathcal{X}$. We assume that $F_k(\zeta_0|x) = 0$ and $F_k(\zeta_B|x) = 1$ for all $k \in [K]$ and $x \in \mathcal{X}$. While survival analysis is often designed to estimate the survival function, defined as $S_k(t|x) = 1 - F_k(t|x)$, this study aims to estimate the cumulative distribution function (CDF) $F_k(t|x)$ of $T_k$ unless stated otherwise. Additionally, this study also considers the estimation of the average CDF $F_k(t)$ of $F_k(t|x)$ over $x \sim \mathcal{X}$ and the average survival function $S_k(t) = 1 - F_k(t)$.

**Censored Joint Distribution (CJD) Representation.** The observation $(x, t, \delta)$ becomes more intuitive when visualized in $K$-dimensional space. For example, in the case where $K = 2$, each observation can be depicted as a line segment in a two-dimensional plane, as illustrated in Figure 2(a). In this figure, an observation $(x^{(1)}, 20, 1)$ is represented as a vertical line segment. This observation indicates that Event 1 is observed at time $t_1 = 20$, and it is only known that $t_2 \geq t_1$. Similarly, another observation $(x^{(2)}, 35, 2)$ is depicted as a horizontal line segment.

Given that the time horizon is discretized, the $K$-dimensional space is partitioned by defining the set $R_{b,k}$ of realizations $(t, \delta) \sim (T, \Delta)$ as follows:

$$R_{b,k} = \{(t, \delta) : \zeta_b \leq t < \zeta_{b+1}, \delta = k\}.$$

Refer to Figure 2(b) for an illustration. In this study, this partitioned region is referred to as the Censored Joint Distribution (CJD) representation.

**Copula and Survival Copula.** In probability theory and statistics, a mathematical construct known as a *copula* is defined as a multivariate cumulative distribution function wherein each variable's marginal probability distribution is uniformly distributed over the interval $[0, 1]$. The primary utility of copulas is to delineate the interdependencies among random variables.

Formally, a copula, according to Nelsen (2006), is a $K$-dimensional function $C : [0, 1]^K \to [0, 1]$ that satisfies the following conditions: (i) $C(u_1, u_2, \ldots, u_{k-1}, 0, u_{k+1}, \ldots, u_K) = 0$, (ii) $C(1, \ldots, 1, u, 1, \ldots, 1) = u$ for every $u \in [0, 1]$, and (iii) given $u, v \in [0, 1]^K$ such that $u_k < v_k$ is valid for all $k \in [K]$, the following condition is satisfied:

$$\sum_{l \in \{0,1\}^K} (-1)^{l_1 + l_2 + \cdots + l_K} C(u_1^{l_1} v_1^{1-l_1}, u_2^{l_2} v_2^{1-l_2}, \ldots, u_K^{l_K} v_K^{1-l_K}) \geq 0.$$

A notable instance of a copula is the *independence copula*, which is expressed as:

$$C_{\text{ind}}(u_1, u_2, \ldots, u_K) = \prod_{k=1}^K u_k. \tag{1}$$

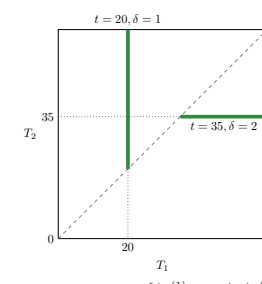 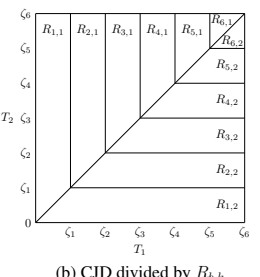 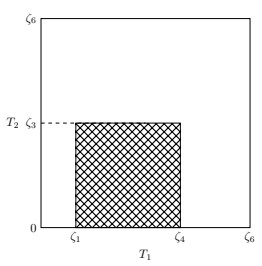

(a) Two observations $\{(x^{(1)}, 20, 1), (x^{(2)}, 35, 2)\}$     (b) CJD divided by $R_{b,k}$     (c) Region specified in $\Pr(\zeta_1 < T_1 \le \zeta_4, T_2 \le \zeta_3)$

Figure 2: Censored Joint Distribution (CJD) Representation: (a) Two observations are illustrated as vertical and horizontal line segments in the CJD representation. (b) The CJD space is divided into subregions $R_{b,k}$ for density estimation. (c) A rectangular region is illustrated within the CJD space.

Another illustration of a bivariate copula is the *Frank copula* with a non-zero parameter $\theta$:

$$C_{\text{Frank}}(u_1, u_2) = \frac{1}{\theta} \log \left( 1 + \frac{(e^{-\theta u_1} - 1)(e^{-\theta u_2} - 1)}{e^{-\theta} - 1} \right). \tag{2}$$

Copulas are instrumental in calculating joint probabilities. For instance, the joint probability $\Pr(\zeta_1 < T_1 \le \zeta_4, T_2 \le \zeta_3)$ depicted in Fig. 2(c) can be computed using a copula as follows:

$$\Pr(\zeta_1 \le T_1 < \zeta_4, T_2 < \zeta_3) = C(F_1(\zeta_4), F_2(\zeta_3)) - C(F_1(\zeta_1), F_2(\zeta_3)),$$

where $F_k(t) = \Pr(T_k \le t)$.

A significant characteristic of the copula is that the joint distribution $\Pr(T_1 \le t_1, T_2 \le t_2, \ldots, T_K \le t_K)$ can be uniquely represented using the copula, as per Sklar's theorem.

**Theorem 1.** *(Sklar's Theorem [1959]). There exists a copula $C$ such that for all $t_1, t_2, \ldots, t_K$,*

$$\Pr(T_1 \le t_1, T_2 \le t_2, \ldots, T_K \le t_K) = C(F_1(t_1), F_2(t_2), \ldots, F_K(t_K)).$$

*If the marginal distribution $F_k$ is continuous for all $k$, then $C$ is unique.*

In the context of survival analysis, a *survival copula* $\overline{C}$ is also frequently used. This copula satisfies the following equation:

$$\Pr(T_1 > t_1, T_2 > t_2, \ldots, T_K > t_K) = \overline{C}(1 - F_1(t_1), 1 - F_2(t_2), \ldots, 1 - F_K(t_K)).$$

It is well-established that any survival copula $\overline{C}$ can be represented using its corresponding copula $C$ (see, e.g., (Georges et al., 2001)). For instance, when $K = 2$, the survival copula $\overline{C}$ can be represented as:

$$\overline{C}(u_1, u_2) = u_1 + u_2 - 1 + C(1 - u_1, 1 - u_2)$$

Additionally, it is important to note that if $C = C_{\text{ind}}$, then $\overline{C} = C_{\text{ind}}$.

## 3 TWO-STEP ALGORITHM

We propose a two-step algorithm for survival analysis with $K$ competing risks. The first step estimates $\hat{r}_{b,k|x}$, which approximates the ground truth $r_{b,k|x} = \Pr((t, \delta) \in R_{b,k}|x)$. In the second step, the marginal distribution $\hat{F}_k(\zeta_b|x)$ is computed from the estimation $\hat{r}_{b,k|x}$, assuming that we have prior knowledge regarding the dependencies among the random variables $T_1, T_2, \ldots, T_K$ in the form of a copula $C$. Note that survival models based on the conditional independence assumption are equivalent to using the independence copula (as defined in (1)) for the copula $C$.

### 3.1 STEP 1: CENSORED JOINT DISTRIBUTION ESTIMATION

We present two approaches to estimate $\hat{r}_{b,k|x}$. One approach is based on density estimation, and the other on distribution regression.

**Density Estimation.** The most straightforward approach is to directly use a density estimation model to estimate $r_{b,k|x}$. Various density estimation models are applicable, including random forests, gradient boosting, and neural network models. Recent advancements in density estimation techniques can be found in (Dheur & Taieb, 2023; Filho et al., 2023).

**Distribution Regression.** Another approach is to utilize distribution regression models. Examples of such models include:

- Models based on monotone neural networks (Chilinski & Silva, 2020)
- Models based on random forests (Schlosser et al., 2019; Hothorn & Zeileis, 2021; Ćevid et al., 2022)
- NGBoost (Duan et al., 2020), which is based on gradient boosting.

These models can estimate $V_k(\zeta|x)$, the conditional $k$-th cumulative incidence function (CIF), defined as follows:

$$V_k(\zeta|x) = \Pr(T \le \zeta, \Delta = k|x). \tag{3}$$

Using the estimated $\hat{V}_k(\zeta|x)$, $\hat{r}_{b,k|x}$ can be estimated as:

$$\hat{r}_{b,k|x} = \hat{V}_k(\zeta_b|x) - \hat{V}_k(\zeta_{b-1}|x). \tag{4}$$

One significant advantage of this approach is that, if we wish to adjust the hyperparameter $B$, there is no need to retrain the predictive model; we only need to recompute Eq. (4).

## 3.2 STEP 2: COMPUTATION OF MARGINAL DISTRIBUTION

The second step of our algorithm computes $\hat{F}_k(\zeta_b|x)$ using the estimates $\hat{r}_{b,k|x}$ obtained in the first step and a given copula $C$. For simplicity, we consider the case for $K = 2$ in this section, with generalization for $K > 2$ detailed in Appendix C.

Let $\mathbf{r}_{b|x} \in [0,1]^K$ denote the length-$K$ vector whose $k$-th entry is $r_{b,k|x}$, and let $\mathbf{F}_{b|x} \in \mathbb{R}^K$ denote the length-$K$ vector whose $k$-th entry is $F_k(\zeta_b|x)$.

We aim to represent $\mathbf{r}_{b|x}$ as a function of $\mathbf{F}_{b-1|x}$, $\mathbf{F}_{b|x}$, and the copula $C$. By the definition of $r_{b,k|x}$, we have the following representations:

$$r_{b,1|x} = \Pr(\zeta_{b-1} < T_1 \le \zeta_b, T_1 \le T_2|x) = q_{\{1\},b|x} - w_1\, q_{\{1,2\},b|x}, \tag{5}$$

$$r_{b,2|x} = \Pr(\zeta_{b-1} < T_2 \le \zeta_b, T_2 \le T_1|x) = q_{\{2\},b|x} - w_2\, q_{\{1,2\},b|x}, \tag{6}$$

where

$$q_{\{1\},b|x} = \Pr(\zeta_{b-1} < T_1 \le \zeta_b, \zeta_{b-1} < T_2|x), \tag{7}$$

$$q_{\{2\},b|x} = \Pr(\zeta_{b-1} < T_2 \le \zeta_b, \zeta_{b-1} < T_1|x), \tag{8}$$

$$q_{\{1,2\},b|x} = \Pr(\zeta_{b-1} < T_1 \le \zeta_b, \zeta_{b-1} < T_2 \le \zeta_b|x), \tag{9}$$

and $w_1, w_2 \ge 0$ are weight parameters such that $w_1 + w_2 = 1$. See Fig. 3 for the illustration of equation 7–equation 8. Unless otherwise stated, we assume that $w_1 = w_2 = 1/2$. Note that, if we use a sufficiently large $B$, the correction term 9 should be a small value, and therefore the choices of the weight parameters $w_1$ and $w_2$ should have little effect in these equations. Then we represent equation 7–equation 9 by using $F_1(\zeta_{b-1}|x), F_1(\zeta_b|x), F_2(\zeta_{b-1}|x), F_2(\zeta_b|x)$, and the copula $C$:

$$q_{\{1\},b|x} = F_1(\zeta_b|x) - F_1(\zeta_{b-1}|x) - C(F_1(\zeta_b|x), F_2(\zeta_{b-1}|x)) + C(F_1(\zeta_{b-1}|x), F_2(\zeta_{b-1}|x))$$

$$= C(F_1(\zeta_b|x), 1) - C(F_1(\zeta_{b-1}|x), 1) - C(F_1(\zeta_b), F_2(\zeta_{b-1}|x))$$
$$+ C(F_1(\zeta_{b-1}|x), F_2(\zeta_{b-1}|x)), \tag{10}$$

$$q_{\{2\},b|x} = C(1, F_2(\zeta_b|x)) - C(1, F_2(\zeta_{b-1}|x)) - C(F_1(\zeta_{b-1}|x), F_2(\zeta_b|x))$$
$$+ C(F_1(\zeta_{b-1}|x), F_2(\zeta_{b-1}|x)), \tag{11}$$

$$q_{\{1,2\},b|x} = C(F_1(\zeta_b|x), F_2(\zeta_b|x)) - C(F_1(\zeta_b|x), F_2(\zeta_{b-1}|x)) - C(F_1(\zeta_{b-1}|x), F_2(\zeta_b|x))$$
$$+ C(F_1(\zeta_{b-1}|x), F_2(\zeta_{b-1}|x)). \tag{12}$$

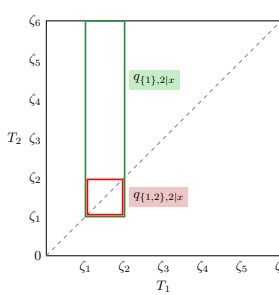

Figure 3: Illustration of $q_{\{1\},2|x}$ and $q_{\{1,2\},2|x}$.

**Algorithm 1** Two-Step Algorithm

---

**Input:** Dataset $\mathcal{D} = \{(x^{(i)}, t^{(i)}, \delta^{(i)})\}_{i=1}^N$.
**Output:** Estimated survival function $\{\hat{\mathbf{F}}_{b|x}\}_{b=1}^{B-1}$.
 1: // Step 1
 2: Estimate $\hat{\mathbf{r}}_{b,k|x}$
 3: // Step 2
 4: Let $\hat{\mathbf{F}}_{0|x} = \mathbf{0}$
 5: **for** $b = 1, 2, \ldots, B-1$ **do**
 6:     Calculate $\hat{\mathbf{F}}_{b|x}$ by solving Eq. (13)
 7: **end for**
 8: **return** $\{\hat{\mathbf{F}}_{b|x}\}_{b=1}^{B-1}$

---

Combining equation 5–equation 12, we can represent $r_{b,1|x}$ and $r_{b,2|x}$ using $F_1(\zeta_{b-1}|x)$, $F_1(\zeta_b|x)$, $F_2(\zeta_{b-1}|x)$, $F_2(\zeta_b|x)$, and the copula $C$. This implies the existence of a function $g_{b,C}$ such that:

$$\mathbf{r}_{b|x} = g_{b,C}(\mathbf{F}_{b|x}|\mathbf{F}_{b-1|x}). \tag{13}$$

Having established equation 13, we can obtain $\mathbf{F}_{b|x}$ for all $b$ by solving this equation, as outlined in Steps 4-7 of Algorithm 1. We leverage the initial condition $\mathbf{F}_{0|x} = \mathbf{0}$ for all $x$, where $\mathbf{0}$ is the $K$-dimensional vector of zeros. At the initial step for $b = 1$, we can obtain $\hat{\mathbf{F}}_{1|x}$ by solving equation 13, since this equation provides $K$ equality constraints and the unknown value is the length-$K$ vector $\mathbf{F}_{b|x}$ (see Sec. C.2 for more details). By repeating this procedure for $b = 2, 3, \ldots, B-1$, we can obtain $\hat{\mathbf{F}}_{b|x}$ for all $b$.

Note that the second step of our algorithm is similar to the algorithm based on a bisection root-finding algorithm in (Zheng & Klein, 1995), but their algorithm is valid only for $K = 2$ and its extension for $K > 2$ is unknown. In contrast, our algorithm is extendable for $K > 2$ as shown in Appendix C.

**Remarks.** In our implementation, we employed a simpler algorithm instead of solving (13) for each $b$. Specifically, we estimate $\hat{\mathbf{F}}_{b|x}$ by minimizing the following objective function:

$$\sum_{b=1}^{B-1} (g_{b,C}(\mathbf{F}_{b|x}|\mathbf{F}_{b|x}) - \hat{\mathbf{r}}_{b|x})^2$$

for all $b$ simultaneously.

We also note that as $B \to \infty$, another approach introduced in (Carrière, 1995) can be employed to estimate $\hat{\mathbf{F}}_{b|x}$. This method is discussed further in Sec. C.3 and we conducted an experimental comparison between this alternative algorithm and our proposed algorithm in Sec. H.

## 4 THEORETICAL ANALYSIS

In this section, we theoretically verify that the two-step algorithm outputs solutions with sufficiently small errors. We consider the case $K = 2$ for simplicity, and we assume $\zeta_b = \frac{b}{B}\zeta_B$. As discussed in the preceding section, various models can be implemented to estimate the CIF in the first step of our algorithm. Therefore, we evaluate errors affected by step 2, solving (13), under the assumption that the models employed in the first step accurately approximate the true probabilities such that

$$|\hat{r}_{b,k} - r_{b,k}| \le \epsilon \tag{14}$$

holds for all $b = 1, \ldots, B$ and $k = 1, 2$. Note that while how small a value we can take as $\epsilon$ in (14) depends on the choice of the model in step 1, we can apply the results exhibited in this section. We provide examples of achieving (14) in Appendix B. To formally state our theoretical results, we introduce the following assumption:

**Assumption 1.** *We assume the following conditions:*

*(1) (True probability is not biased.) There exists a global constant $c_0 > 0$ such that for every $b = 1, \ldots, B$ and $k = 1, 2$, $F_k(\zeta_b|x) - F_k(\zeta_{b-1}|x) = \Pr(\zeta_{b-1} < T_k \leq \zeta_b) \leq \frac{c_0}{B}$ holds.*

*(2) (Copula.) We assume that the copula $C$ is of class $\mathcal{C}^2$ and satisfies*

$$\ell := \inf_{(u,v) \in [0,1]^2} \frac{\partial^2}{\partial u \partial v} C(u,v) > 0,$$

$$L := \sup_{(u,v) \in [0,1]^2} \max\left\{ \frac{\partial^2}{\partial u^2} C(u,v), \frac{\partial^2}{\partial u \partial v} C(u,v), \frac{\partial^2}{\partial v^2} C(u,v) \right\} < +\infty.$$

*(3) (All $t_k$ are equally observed.) There exist constants $c_1 > 0$ depend on $\ell$, $L$, and $\tau > 0$ such that the following condition holds: Let $\delta_0 > 0$ be a constant determined by $\ell$ and $L$ [1] and $b_0 := \max_b \left\{ b \mid \forall k, F_k(\zeta_b|x) \leq 1 - \frac{\delta_0}{\tau \log B} \right\}$. Then, $\min_k F_k(\zeta_{b_0}|x) \geq 1 - \frac{c_1}{\log B}$.*

We make some remarks on the assumption. The first condition is required to bound the error by the choice of $w_1$ and $w_2$; if the probability concentrates on a squared region partitioned by suboptimal $w_1$ and $w_2$, significant errors are inevitable. This condition is satisfied if $F_k$ is Lipschitz-continuous with $c_0 \zeta_B$ serving as the Lipschitz constant. The second condition manages the sensitivity of the estimation relative to the true distribution and noise. For example, if $\frac{\partial^2 C}{\partial u \partial v} \ll 1$, indicating that $C$ exhibits minimal variation as $u$ and $v$ change, substantial adjustments to the estimation are necessary to accommodate for noise and achieve (13). This condition is typically met for the independence copula $C(u,v) = uv$ with any $\ell < 1$ and $L > 1$. The third condition appears to be technical. As will be demonstrated in subsequent analyses, errors between $\hat{F}_k$ and $F_k$ can only be effectively bounded for $b \leq b_0$. As $b$ approaches $B$, and consequently $r_{b,k}$ diminishes, the impact of $\epsilon$ intensifies. Condition (3) excludes scenarios where a part of $t_k$s is concentrated in the region $b > b_0$. In other words, all $t_k$s are equally observed in the region $b \leq b_0$.

Let $W_1(\cdot, \cdot)$ be the Wasserstein distance[2]. Then, we provide the statement about the $W_1$ distance between between the estimated and true probabilities. We consider the extension of $\hat{F}_k(\zeta_b|x)$ to a CDF on $[0, \zeta_B]$ by $\hat{F}_k(t|x) := \hat{F}_k(\zeta_b|x)$, where $\zeta_b \leq t < \zeta_{b+1}$.

**Theorem 2.** *Suppose that Assumption 1 holds. Then, there exists a constant $c_\epsilon > 0$ depending $c_0$, $\ell$ and $L$ such that if $\epsilon \leq \frac{c_\epsilon}{B}$, the following inequality holds:*

$$W_1\left(\hat{\mu}_{k|x}, \mu_{k|x}\right) \lesssim \zeta_B \left( B^{1+\tau} \epsilon + c_1 \cdot \frac{B - b_0}{B \log B} \right), \tag{15}$$

*where $\hat{\mu}_{k|x}$ and $\mu_{k|x}$ are probability measures whose CDFs are given by $\hat{F}_k(\cdot|x)$ and $F_k(\cdot|x)$, respectively.*

Due to the space limitation, the proof is deferred to Appendix D. Suppose that the condition $\epsilon = o(B^{1+\tau})$ holds. Then, we obtain an upper bound as $W_1\left(\hat{\mu}_{k|x}, \mu_{k|x}\right) = \zeta_B \cdot o(1)$. Thus, we can ensure that as $B \to +\infty$ and the sample size increases as we can take sufficiently small $\epsilon$, the output of step 2 converges to the ground truth distribution in terms of the $W_1$ distance.

We provide some comments on Theorem 2. We can observe a trade-off in (15) based on the choice of $B$: while the second term decreases as $B$ increases, $B^{1+\tau}$ and $\epsilon$ in the first term should increase. Consequently, an optimal choice of $B$ should be considered under appropriate assumptions that determine $\epsilon$ and $b_0$, such as the model utilized in step 1 and the properties of $\mathbf{F}_{\cdot|x}$. It is also significant to examine that the derived bound achieves a statistical min-max lower bound exhibited in (Niles-Weed & Berthet, 2022; Bilodeau et al., 2023), for example. We reserve these considerations for future research endeavors.

---

[1]See Section D for its formal definition.

[2]$W_1(\mu, \nu) := \inf_{\pi \in \Pi(\mu,\nu)} \int_{\mathbb{R}^2} |x - y| \mathrm{d}\pi(x,y)$, where $\Pi(\mu,\nu)$ denotes the set of all couplings of two probability measures $\mu$ and $\nu$ on $\mathbb{R}$.

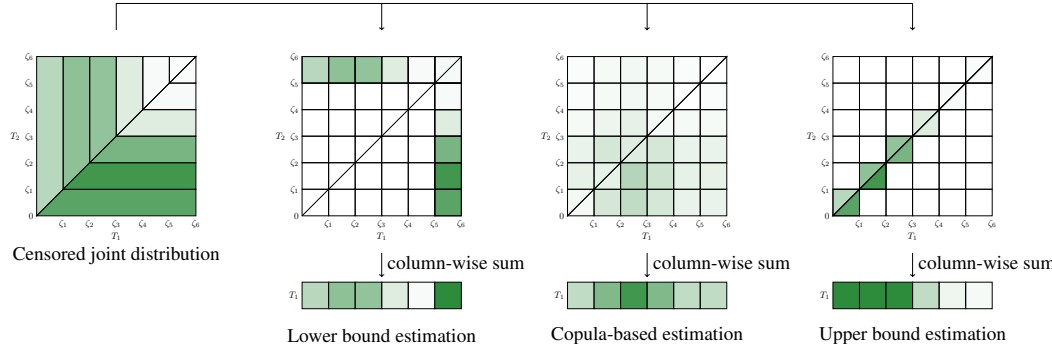

Figure 4: Illustration of the upper and lower bounds estimation. Here, region $R_{b,k}$ is divided into grids, with denser color indicating a higher probability that a data point is contained in the corresponding region. The lower bound estimation is achieved by assigning the probability mass $\hat{r}_{b,k}$ to the last time slot within region $R_{b,k}$ and then calculating the column-wise sum. Conversely, the upper bound estimation is obtained by assigning the probability mass $\hat{r}_{b,k}$ to the earliest time slot within region $R_{b,k}$ and calculating the column-wise sum.

## 5 UPPER AND LOWER BOUNDS ESTIMATION

While our two-step algorithm assumes prior knowledge of the copula $C$, Tsiatis (1975) demonstrates that further weakening this assumption renders it impossible to identify $F_k(t|x)$. For additional details, see our discussion in Sec. E.

Given this constraint, we explore the concept of partial identifiability (Kline & Tamer, 2023) in survival analysis. Specifically, we derive the upper and lower bounds of $F_k(t|x)$ under the assumption that the true copula $C$ is unknown. By definition, the upper and lower bounds of $F_k(\zeta_b|x)$ can be computed as follows:

$$\Pr\left((x,t,\delta) \in \bigcup_{b' \leq b} R_{b',k|x}\right) \leq F_k(\zeta_b|x) \leq \Pr\left((x,t,\delta) \in \bigcup_{b' \leq b, k' \in [K]} R_{b',k'|x}\right)$$

$$\Leftrightarrow \qquad \sum_{b' \leq b} r_{b',k|x} \leq F_k(\zeta_b|x) \leq \sum_{b' \leq b, k' \in [K]} r_{b',k'|x}. \tag{16}$$

Note that these inequalities are derived without utilizing the parameters $w_1$ and $w_2$ in our two-step algorithm.

Given (16), we can compute the upper and lower bounds using $\hat{r}_{b,k|x}$, as demonstrated in Figure 4. As illustrated, the upper and lower bound estimation, as well as our two-step algorithm to estimate $F_k(\zeta_b|x)$, can be interpreted as redistributing the probability mass in the CJD representation into fine-grained grid cells.

It is important to note that our upper and lower bounds differ from the *confidence interval*, which quantifies the epistemic uncertainty inherent to the prediction model (Bengs et al., 2022). Our bounds quantify uncertainty due to the lack of prior knowledge about the true copula $C$. As discussed in Sec. F, these two bounds can be combined to quantify both uncertainties.

In Sec. F, we present alternative upper and lower bounds using $V_k(\zeta|x)$ (as defined in (3)). These bounds can be considered variants of the bounds given for the average survival function in (Peterson, 1976).

## 6 STRICTLY PROPER SCORING RULE

In this section, we demonstrate the existence of a strictly proper scoring rule when the ground truth (survival) copula is known. While the existence of such a scoring rule for $K = 2$ under the

conditional independence assumption is shown in (Rindt et al., 2022), no strictly proper scoring rule has been established under our weaker assumption.

**Definition 1.** (Proper and strictly proper scoring rules.) A scoring rule $\mathcal{S}$ for estimation $\hat{F}_k(t|x)$ of $F_k(t|x)$ is *proper* if the following inequality holds:

$$\mathbb{E}[\mathcal{S}(\{\hat{F}_k(t|x)\}_{k=1}^K, (t, \delta))] \geq \mathbb{E}[\mathcal{S}(\{F_k(t|x)\}_{k=1}^K, (t, \delta))]. \tag{17}$$

A scoring rule is *strictly proper* if the equality in (17) holds if and only if $\hat{F}_k(t|x)$ is equal to $F_k(t|x)$ for all $k$ and $t$.

To define our scoring rule, let

$$v_k(t|x) = \frac{d}{dt} V_k(t|x)$$

$$= -\left.\frac{\partial}{\partial t_k} \overline{C}(1 - F_1(t_1|x), 1 - F_2(t_2|x), \ldots, 1 - F_K(t_K|x))\right|_{t_1 = t_2 = \cdots = t_K = t}.$$

**Assumption 2.** *An estimate $\hat{v}_k(t|x)$ of $v_k(t|x)$ satisfies $D_{\mathrm{KL}}(v_k(t|x)||\hat{v}_k(t|x)) < \infty$ for all $k$ and $t$, where $D_{\mathrm{KL}}$ denotes the Kullback-Leibler (KL) divergence.*

**Theorem 3.** *(Strictly proper scoring rule.) If Assumption 2 holds, the following scoring rule $\mathcal{S}$, termed Copula-NLL, is strictly proper:*

$$\mathcal{S}(\{\hat{F}_k(t|x)\}_{k=1}^K, (t, \delta)) = -\mathbb{1}_{\delta=k} \log \hat{v}_k(t|x).$$

In Sec. G, we provide the proof of this theorem.

# 7 EXPERIMENTS

We conducted experimental evaluations to verify that our two-step algorithm delivers comparable predictive performance to existing models for survival analysis. Additionally, we evaluated a neural network model utilizing our strictly proper scoring rule, Copula-NLL. For this section, we used two datasets: flchain (Kyle et al., 2006; Dispenzieri et al., 2012) and support2 (Knaus et al., 1995), both obtained from the Python package `SurvSet` (Drysdale, 2022).

**Models.** We employed four models for the first step of our algorithm. In the models TS-Brier and TS-Log, neural networks were used for density estimation with the Brier and Logarithmic scores (Gneiting & Raftery, 2007) as the loss functions, respectively. In the models TS-RF and TS-DRF, we used the random forest (RF) model provided in the `sklearn` package and the distribution regression forest (DRF) (Ćevid et al., 2022), respectively. The prefix TS stands for the Two-Step algorithm common to these models, each employing a different first-step model, but all using the same algorithm for the second step as described in Algorithm 1. We set $B = 100$ for the hyperparameter of the second step.

Additionally, we constructed a Copula-NLL model that used our strictly proper scoring rule presented in Sec. 6 as its loss function. This model utilized a min-max neural network (Igel, 2024) to represent a CDF with a *monotone* neural network (Chilinski & Silva, 2020), though other monotone neural networks such as those proposed in (Yanagisawa et al., 2022; Kim & Lee, 2024) could also be used. The Copula-NLL model can be seen as incorporating prior knowledge of the ground truth copula into the DCSurvival model (Zhang et al., 2024), although the primary objective of DCSurvival appears to be the identification of parameters within an Archimedean copula.

For comparison, we also employed the random survival forest (RSF) (Ishwaran et al., 2008), a model based on random forests, and the DeepHit model (Lee et al., 2018), a neural network model. In the DeepHit model, we set the parameter $\alpha = 0$ to make its loss function a proper scoring rule as suggested by Yanagisawa (2023).

**Evaluation Metrics.** We used three types of metrics to evaluate the estimate on the CJD representation $\hat{r}_{b,k|x}$ and four types of metrics to evaluate the estimated distribution $\hat{F}_k(t|x)$ (see also Table 1 in Appendix H).

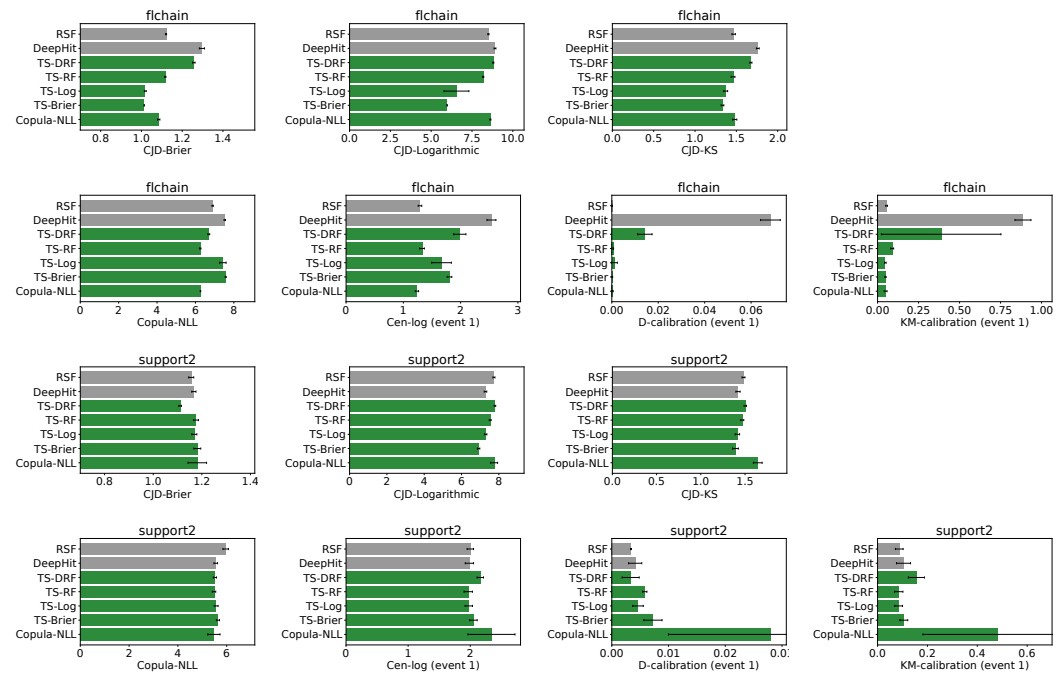

Figure 5: Performance comparison on the flchain and support2 datasets (lower is better).

For the CJD representation, we employed the Brier and Logarithmic scores (Gneiting & Raftery, 2007). Given that certain models yield only the marginal distribution $\hat{F}_k(t|x)$, the CJD representation $\hat{\mathbf{r}}_b$ was estimated using the marginal distribution $\hat{F}_k(t|x)$ and the independence copula $C_{\text{ind}}$ (defined in (1)). Additionally, the sum of the Kolmogorov-Smirnov calibration error was used as a calibration metric on the CJD representation. This metric, used in (Gupta et al., 2021), is based on the Kolmogorov-Smirnov test (Kolmogorov, 1933; Smirnov, 1939) and is defined as follows:

$$\sum_{k=1}^{K} \max_{0 \leq \sigma \leq 1} \left| \frac{1}{N} \sum_{i=1}^{N} \mathbb{1}_{f_k(x_i) \leq \sigma} \cdot \mathbb{1}_{y_i=k} - \frac{1}{N} \sum_{i=1}^{N} \mathbb{1}_{f_k(x_i) \leq \sigma} \cdot f_k(x_i) \right|,$$

where $f_k(x_i)$ denotes the probability of being $x_i$ classified in class $k$.

For the estimated marginal distribution $\hat{F}_k(t|x)$, we used a simplified variant of the censored logarithmic score (Yanagisawa, 2023) as the evaluation metric. Additionally, D-calibration (Haider et al., 2020) and KM-calibration (Yanagisawa, 2023) were employed as calibration metrics. Finally, we used our strictly proper scoring rule, Copula-NLL, as an evaluation metric.

**Results.** Figure 5 presents the results for the flchain and support2 datasets. These results indicate that no single model consistently outperforms all others across different metrics. Our models generally exhibited competitive or superior predictive performance compared to RSF and DeepHit models.

The TS-DRF, TS-RF, TS-Log, and TS-Brier models' values on the CJD-Brier, CJD-Logarithmic, and CJD-KS metrics reflect the prediction performances of the density estimation methods used in the first step of the two-step algorithm. This suggests that no single prediction model uniformly outperforms the others, even in terms of density estimation. Given that each prediction model has its own inductive bias, the "best model" is contingent on the dataset and the specific evaluation metric. We defer the problem of model selection to existing works (e.g., (Arlot & Celisse, 2010; Arlot & Lerasle, 2016)).

Additional evaluation results using several other datasets are included in Appendix H.

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

## A    ON APPLICATIONS OF PROPOSED METHODS

Survival analysis is a crucial tool in various fields such as medicine, engineering, and social sciences, where the time until an event of interest occurs is studied. The applicability of our proposed methods can be classified into three distinct types based on the nature of the dependency between the event time and the censoring time.

- (Conditional Independence.) The first class of applications encompasses scenarios where the conditional independence assumption between the event time and the censoring time is valid or highly likely to hold. This situation typically arises in cases of administrative censoring, where data points are censored solely due to the limited window of observation times. For instance, in clinical trials, patients might be censored at the end of the study period regardless of whether the event of interest has occurred. In such cases, the conditional independence assumption simplifies the analysis, allowing the use of survival models based on the conditional independence assumption including our two-step algorithm.

- (Verifiable Dependency.) The second class of applications includes scenarios where there is a dependency between the event time and the censoring time, but this dependency can be verified, albeit at a significant cost, for a small subset of data points. An example of this situation is found in medical studies where the primary event of interest is patient mortality, and censoring occurs when patients are discharged from the hospital. In such cases, it might be feasible to investigate the true event time for a small fraction of patients, thereby assessing the dependency between the event and censoring times. For these applications, our two-step algorithm can be adapted by estimating the copula $C$ for the subset of data points where the dependency has been verified. This estimated copula can then be used to model the dependency in the entire dataset, allowing for more accurate estimation of individual survival functions.

- (Unverifiable Dependency.) The third class of applications involves situations where the dependency between the event time and the censoring time cannot be determined, even with extensive resources. In such scenarios, identifying the individual survival function is inherently challenging due to the unknown nature of the dependency. Without precise knowledge of the dependency, it is prudent to consider a range of potential dependencies. For example, if we have some confidence that the dependency can be represented by a Frank copula with parameters $-5 \leq \theta \leq 5$, we can estimate the survival functions using the Frank copula with $\theta = -5$ and $\theta = 5$. These estimates provide the upper and lower bounds of the true survival function. Additionally, our method for estimating the upper and lower bounds offers a robust approach to account for the uncertainty in the dependency structure.

## B    EXAMPLES OF BOUNDS ON STEP 1 ERROR

In this section, we present an example that satisfies (14). As mentioned in Sections 3 and 4, any estimation method for the probability distribution can be employed, and theoretical results pertaining to those models can be leveraged to guarantee (14). Among the various methods, we introduce results derived from the Distributional Random Forest (DRF) (Ćevid et al., 2022) and histogram type estimators (Sart, 2017).

### B.1    DISTRIBUTIONAL RANDOM FOREST

DRF constructs random forests designed to estimate the conditional distribution of multivariate responses. It achieves this by splitting the data using a distributional metric, specifically the maximal mean discrepancy (MMD), with the goal of maximizing the differences in distributions between child nodes. DRF then estimates targets, such as the CIF in this study, by employing a weight function that reflects how frequently the training data points end up in the same leaf as the test point across different trees.

Ćevid et al. (2022) imposes the following assumptions:

(P1) *(Data Sampling.)* Instead of the traditional bootstrap sampling with replacement, commonly used in forest-based methods, a subsampling approach is employed. For each tree, a random subset of size $s_n$ is selected from $n$ training data points. It is assumed that $s_n$ approaches infinity as $n$ increases, with the rate specified below.

(P2) *(Honesty.)* The data used to construct each tree is split into two parts: one part is used for determining the splits, and the other is used for populating the leaves and thus for estimating the response.

(P3) *($\alpha$-Regularity.)* Each split leaves at least a fraction $0 < \alpha \le 0.2$ of the available training sample on each side. Additionally, trees are grown until each leaf contains between $\kappa$ and $2\kappa - 1$ observations, where $\kappa \in \mathbb{N}$ is a fixed tuning parameter.

(P4) *(Symmetry.)* The (randomized) output of a tree does not depend on the ordering of the training samples.

(P5) *(Random-Split.)* At every split point, the probability that the split occurs along the feature $X_j$ is bounded below by $\pi/p$, for some $\pi > 0$ and for all $j = 1, \ldots, p$.

Note that each of these conditions can be verified by inspecting the constructed forest. The following proposition is direct consequence from Corollary 5 of Ćevid et al. (2022).

**Proposition 1.** *Under the assumptions (P1)-(P5), it holds that*

$$\hat{r}_{b,k|x} \xrightarrow{p} r_{b,k|x}$$

*for any $b$ and $k$ as the sample size goes to infinity*[3].

The proposition above ensures the probabilistic convergence of the estimation. Specifically, for arbitrary values of $\epsilon > 0$ and $\delta > 0$, there exists a sample size threshold $n_{\epsilon,\delta} > 0$ such that if the sample size exceeds $n_{\epsilon,\delta}$, then (14) holds with a probability of at least $1 - \delta$.

### B.2 HISTOGRAM TYPE ESTIMATORS

Sart (2017) proposes histogram type conditional density estimators on $\mathcal{X} \times \mathcal{Y}$ by

$$\widehat{\Pr}(y|x) \coloneqq \hat{s}(x,y) = \sum_{K \in m} \frac{\sum_{i=1}^{n} \mathbb{1}_K(x_i, y_i)}{\sum_{i=1}^{n} (\delta_{x_i} \otimes \mu)(K)} \mathbb{1}_K(x,y),$$

where $m$ is a partition of $\mathcal{X} \times \mathcal{Y}$, $\mu$ is a reference measure of the conditional density, and $\delta_x$ is the Dirac measure at $x \in \mathcal{X}$. In the context of survival analysis, we can choose $\mathcal{Y} = [0,1]^K$ equipped with the Lebesgue measure $\mu$ and $m$ as a set of regions defined by $(\zeta_{b-1} \le t_k < \delta_b, \delta = k)$ for $b \in [B], k \in [K]$.

Let $\nu$ be a measure defined on $\mathcal{X}$ and

$$h(f,g) = \int_{\mathcal{X} \times \mathcal{Y}} \left( \sqrt{f(x,y)} - \sqrt{g(x,y)} \right)^2 \mathrm{d}\nu(x) \, \mathrm{d}\mu(y)$$

be the Hellinger distance. Then, Sart (2017) provides the following result:

**Proposition 2** (Proposition 2.6 of Sart (2017)). *Let $s$ be a true conditional density. We define*

$$V_m \coloneqq \left\{ \sum_{K \in m} a_K \mathbb{1}_K, \forall K \in m, a_K \ge 0 \right\}.$$

*Then, there exists global constants $C_1$, $C_2 > 0$ such that for any $\xi > 0$,*

$$\Pr\left[ h^2(s, \hat{s}) \le \inf_{v \in V_m} h^2(s, v) + C_1 \frac{|m|}{n} + C_2 \xi \right] \ge 1 - e^{-n\xi}.$$

See Sart (2017) for specific examples of deriving the term $\inf_{v \in V_m} h^2(s, v)$ under the conditions on $s$ and $\mathcal{X}$. The bound on the Hellinger distance implies the bound on the total variation distance $\mathrm{TV}(f,g) = \int_{\mathcal{X} \times \mathcal{Y}} |f(x,y) - g(x,y)| \, \mathrm{d}\nu(x) \, \mathrm{d}\mu(y)$ as

$$\mathrm{TV}(f,g) \le h(f,g),$$

which follows from the inequality between the $L_1$-norm and the $L_2$-norm. Thus, by utilizing Proposition 2, we obtain (14) by taking $\epsilon$ as the total variation distance between $\hat{r}$ and $r$.

---

[3] $\xrightarrow{p}$ denotes the probability convergence.

## C    Step 2 of Proposed Algorithm

### C.1    Generalization for competing risks

We generalize the second step of our two-step algorithm presented in Sec. 3 for $K > 2$.

For a subset $\mathbf{I} \subseteq [K]$, let

$$Q_{\mathbf{I},b|x} = \{(t_1, t_2, \ldots, t_K) : \wedge_{k' \in \mathbf{I}}(\zeta_{b-1} < t_{k'} \leq \zeta_b) \text{ and } \wedge_{k' \notin \mathbf{I}} (\zeta_{b-1} \leq t_{k'})\}.$$

We can compute the probability $q_{\mathbf{I},b|x} = \Pr((t_1, t_2, \ldots, t_K) \in Q_{\mathbf{I},b|x})$ by using the inclusion-exclusion principle:

$$q_{\mathbf{I},b|x} = \sum_{j=0}^{K} (-1)^{K-j} \sum_{\mathbf{J}:\mathbf{J} \subseteq [K], |\mathbf{J}|=j} c(\mathbf{I}, \mathbf{J}, b),$$

where $c(\mathbf{I}, \mathbf{J}, b) = C(p_{\mathbf{I},\mathbf{J},b,1}, p_{\mathbf{I},\mathbf{J},b,2}, \ldots, p_{\mathbf{I},\mathbf{J},b,K})$ and

$$p_{\mathbf{I},\mathbf{J},b,k} = \begin{cases} 1 & \text{if } k \in \mathbf{J} \setminus \mathbf{I}, \\ F_k(\zeta_b|x) & \text{if } k \in \mathbf{J} \cap \mathbf{I}, \\ F_k(\zeta_{b-1}|x) & \text{if } k \notin \mathbf{J}. \end{cases}$$

By generalizing Eq. (7)–(9) using $q_{\mathbf{I},b|x}$, we can represent $r_{b,k}$ as

$$r_{b,k|x} = q_{\{k\},b|x} - \underbrace{\sum_{i=2}^{K} (-1)^i \sum_{\mathbf{H}:k \in \mathbf{H} \subseteq [K], |\mathbf{H}|=i} w_{\mathbf{H}} \cdot q_{\mathbf{H},b|x}}_{\text{Correction term}},$$

where $w_{\mathbf{H}}$ is a weight parameter and we assume that $w_{\mathbf{H}} = 1/|\mathbf{H}|$.

Since $q_{\mathbf{I},b|x}$ are functions of $\mathbf{F}_{b|x}$, $\mathbf{F}_{b-1|x}$, and the copula $C$, we can obtain the simultaneous equations:

$$\mathbf{r}_b = g_{b,C}(\mathbf{F}_b|\mathbf{F}_{b-1}).$$

Hence we can estimate $\hat{\mathbf{F}}_{b|x}$ by using Algorithm 1.

### C.2    Solving Simultaneous Equations

In this section, we demonstrate how equation (13) can be solved using Algorithm 2. This algorithm capitalizes on the following property:

**Property 1.** Assuming that $\hat{\mathbf{F}}_{b-1|x}$ is fixed:

(1) The $k$-th element of the length-$K$ vector $g_{b,C}(\hat{\mathbf{F}}_{b|x}|\hat{\mathbf{F}}_{b-1|x})$ is *monotonically increasing* with respect to the $k$-th element of $\hat{\mathbf{F}}_{b|x}$.

(2) The $k'(\neq k)$-th element of the length-$K$ vector $g_{b,C}(\hat{\mathbf{F}}_{b|x}|\hat{\mathbf{F}}_{b-1|x})$ is *monotonically decreasing* with respect to the $k'$-th element of $\hat{\mathbf{F}}_{b|x}$.

First, we demonstrate that the following inequality always holds during the execution of Algorithm 2:

$$\hat{\mathbf{r}}_{b|x} \geq g_{b,C}(\hat{\mathbf{F}}_{b|x}|\hat{\mathbf{F}}_{b-1|x}). \tag{18}$$

At line 1 of Algorithm 2, $\hat{\mathbf{F}}_{b|x}$ is initialized with $\hat{\mathbf{F}}_{b-1|x}$. Hence, by the definition in Eq. (13), we have $g_{b,C}(\hat{\mathbf{F}}_{b|x}|\hat{\mathbf{F}}_{b-1|x}) = \mathbf{0}$, which means that inequality (18) holds. At line 4 of this algorithm, we can increase the $k$-th element of $\hat{\mathbf{F}}_{b|x}$ to satisfy $\hat{\mathbf{r}}_{b|x} = g_{b,C}(\hat{\mathbf{F}}_{b|x}|\hat{\mathbf{F}}_{b-1|x})$ due to Property 1(1), and this increment does not violate inequality (18) due to Property 1(2).

Since each element in the length-$K$ vector $\hat{\mathbf{F}}_{b|x}$ does not decrease during the execution of this algorithm, we can find the solution to equation (13) by repeating the while-loop (lines 2–6) until the convergence of $\hat{\mathbf{F}}_{b|x}$.

---

**Algorithm 2** Algorithm to solve simultaneous equations

---

**Input:** Equation $\hat{\mathbf{r}}_{b|x} = g_{b,C}(\mathbf{F}_{b|x}|\hat{\mathbf{F}}_{b-1|x})$.

**Output:** Solution $\hat{\mathbf{F}}_{b|x}$.

1: Initialize $\hat{\mathbf{F}}_{b|x} = \hat{\mathbf{F}}_{b-1|x}$

2: **while** $\hat{\mathbf{F}}_{b|x}$ is not converged **do**

3:     **for** $k \in \{1, 2, \ldots, K\}$ **do**

4:         Increase the $k$-th element of $\hat{\mathbf{F}}_{b|x}$ so that the $k$-th equation of $\hat{\mathbf{r}}_{b|x} = g_{b,C}(\hat{\mathbf{F}}_{b|x}|\hat{\mathbf{F}}_{b-1|x})$ is

        satisfied (while other $k'$-th element ($k \neq k'$) of $\hat{\mathbf{F}}_{b|x}$ is fixed)

5:     **end for**

6: **end while**

7: **return** $\hat{\mathbf{F}}_{b|x}$

---

### C.3   Alternative Algorithm for Step 2

In the second step of our two-step algorithm, $r_{b,k|x}$ can be represented as $\frac{d}{dt} V_k(t)$ when $B \to \infty$. Therefore, the second step of our two-step algorithm can be seen as solving this partial differential equation:

$$
\begin{aligned}
\frac{d}{dt} V_k(t|x) &= -\left.\frac{\partial}{\partial t_k}\overline{C}(1 - F_1(t_1|x), 1 - F_2(t_2|x), \ldots, 1 - F_K(t_K|x))\right|_{t_1=t_2=\cdots=t_K=t}, \\
&= -\left.\frac{\partial}{\partial t_k}\overline{C}(S_1(t_1|x), S_2(t_2|x), \ldots, S_K(t_K|x))\right|_{t_1=t_2=\cdots=t_K=t}, \\
&= -\left.\frac{\partial\overline{C}(u_1, u_2, \ldots, u_K)}{\partial u_k}\right|_{u_1=S_1(t), u_2=S_2(t), \ldots, u_K=S_K(t)} \left.\frac{d}{dt} S_k(t_k|x)\right|_{t_k=t}, \quad (19)
\end{aligned}
$$

where $\overline{C}$ is the survival copula corresponding to $C$.

In the context of estimating the average survival function $S_k(t) = 1 - F_k(t)$, Equation (19) can be simplified by marginalizing over $x \sim X$ as

$$
\frac{d}{dt} V_k(t) = -\left.\frac{\partial\overline{C}(u_1, u_2, \ldots, u_K)}{\partial u_k}\right|_{u_1=S_1(t), u_2=S_2(t), \ldots, u_K=S_K(t)} \left.\frac{d}{dt} S_k(t_k)\right|_{t_k=t}. \quad (20)
$$

Assuming that we have an estimation $\hat{V}_k(t)$ of $V_k(t)$, Carrière (1995) shows that we can obtain estimation $\hat{S}_k(t)$ for all $k$ by solving Eq. (20). To solve this equation, his algorithm uses these approximations:

$$
u_k \approx (S_k(\zeta_b) + S_k(\zeta_{b+1}))/2 \quad (21)
$$

$$
\left.\frac{d}{dt} S_k(t)\right|_{t=\zeta_b} \approx (S_k(\zeta_{b+1}) - S_k(\zeta_b))/(\zeta_{b+1} - \zeta_b) \quad (22)
$$

$$
\left.\frac{d}{dt} V_k(t)\right|_{t=\zeta_b} \approx (V_k(\zeta_{b+1}) - V_k(\zeta_b))/(\zeta_{b+1} - \zeta_b) \quad (23)
$$

for all $k$.

It is easy to extend this algorithm to solve Eq. (19) by using the variants of Eq. (21), (22), and (23) conditional on $x$. Therefore, we conducted numerical experiments to compare this algorithm with our two-step algorithm in Sec. H.

## D   Proofs

This section provides proofs that remain in Section 4. In this section, for notational simplicity, we abbreviate the conditional variable $x$. For example, we denote $F_k(\zeta_b)$ instead of $F_k(\zeta_b|x)$. We

assume $w_1 = w_2 = \frac{1}{2}$ just for simplicity. We can extend our analysis to arbitrary $w_1$, $w_2 \in (0, 1)$. Moreover, we assume that step 2 exactly solves (13), i.e., $\hat{F}_k$ exactly satisfies the equation (13).

First, we evaluate the error between the outputs of step 2 and the true probability.

**Proposition 3** (Estimation error when solving (13)). *Suppose that Assumption 1 holds. Let $\hat{\mathbf{F}}_{b|x}$ be the output of Algorithm 1. Then, for every $b = 1, \ldots, B$ and $k = 1, 2$, there exists a constant $c_\epsilon > 0$ depending $c_0$, $\ell$ and $L$ such that under the condition $\epsilon \leq \frac{c_\epsilon}{B}$, for sufficiently large $B$ and every $b$ satisfying $\max_{k \in \{1,2\}} F_k(\zeta_b) \leq 1 - \delta$ with a positive constant $\delta$ satisfying $\frac{8c_0 L}{\tau \ell \log B} \leq \delta$ with $\tau \in (0, \frac{1}{2})$,*

$$|\hat{F}_k(\zeta_b) - F_k(\zeta_b)| \leq \left[ \epsilon + \frac{16L\delta^{-2}}{\ell^2} \left( \epsilon + L \cdot \frac{c_0}{B} \right)^2 \right] \cdot \left( \frac{1}{\ell} + \frac{B}{4c_0 L} \right) \left( 1 - \frac{8c_0 L}{B\ell\delta} \right)^{-b+1}. \quad (24)$$

*Proof.* Let us denote $\Delta_{b,k} := \hat{F}_k(\zeta_b) - F_k(\zeta_b)$. Instead of (24), we aim to obtain a tighter bound

$$|\Delta_{b,k}| \leq \left[ \epsilon + \frac{16L\delta^{-2}}{\ell^2} \left( \epsilon + L \cdot \frac{c_0}{B} \right)^2 \right] \cdot \left[ \left( \frac{1}{\ell} + \frac{B}{4c_0 L} \right) \left( 1 - \frac{8c_0 L}{B\ell\delta} \right)^{-b+1} - \frac{B}{4c_0 L} \right]. \quad (25)$$

We give its proof by induction on $b$. For the case $b = 1$, we have

$$C(\hat{F}_1(\zeta_1), 1) = \hat{r}_{1,1} + w_1 \cdot C(\hat{F}_1(\zeta_1), \hat{F}_2(\zeta_1))$$

and

$$C(F_1(\zeta_1), 1) = r_{1,1} + w_{1,1}^* \cdot C(F_1(\zeta_1), F_2(\zeta_1))$$

with $w_{1,1}^* \in [0, 1]$. By taking the difference of the both sides, we obtain

$$\begin{aligned}
&C(\hat{F}_1(\zeta_1), 1) - C(F_1(\zeta_1), 1) \\
&= \hat{r}_{1,1} - r_{1,1} + w_1 \cdot C(\hat{F}_1(\zeta_1), \hat{F}_2(\zeta_1)) - w_{1,1}^* \cdot C(F_1(\zeta_1), F_2(\zeta_1)).
\end{aligned} \quad (26)$$

First, we evaluate the term $C(\hat{F}_1(\zeta_1), \hat{F}_2(\zeta_1))$. Rearranging terms gives

$$\begin{aligned}
&C(\hat{F}_1(\zeta_1), 1) - w_1 \cdot C(\hat{F}_1(\zeta_1), \hat{F}_2(\zeta_1)) \\
&= \hat{r}_{1,1} - r_{1,1} + C(F_1(\zeta_1), 1) - w_{1,1}^* \cdot C(F_1(\zeta_1), F_2(\zeta_1)) \\
&\leq \epsilon + C(F_1(\zeta_1), 1) \\
&\leq \epsilon + L \cdot \frac{c_0}{B},
\end{aligned}$$

where the first inequality follows from $\hat{r}_{1,1} - r_{1,1} \leq \epsilon$ by (14) and $w_{1,1}^* \cdot C(F_1(\zeta_1), F_2(\zeta_1)) \geq 0$, and the last inequality is derived from $F_1(\zeta_1) \leq \frac{c_0}{B}$ by Assumption 1-(1) and

$$C(F_1(\zeta_1), 1) = \int_0^{F_1(\zeta_1)} \int_0^1 \underbrace{\frac{\partial^2}{\partial u \partial v} C(u, v)}_{\leq L} \, \mathrm{d}v \mathrm{d}u \leq L \cdot F_1(\zeta_1) \leq L \cdot \frac{c_0}{B}.$$

Since $C(\hat{F}_1(\zeta_1), \hat{F}_2(\zeta_1)) \leq C(\hat{F}_1(\zeta_1), 1)$, we obtain

$$C(\hat{F}_1(\zeta_1), 1) - w_1 \cdot C(\hat{F}_1(\zeta_1), 1) \leq C(\hat{F}_1(\zeta_1), 1) - w_1 \cdot C(\hat{F}_1(\zeta_1), \hat{F}_2(\zeta_1)) \leq \epsilon + L \cdot \frac{c_0}{B}.$$

By using $1 - w_1 = \frac{1}{2}$, we obtain

$$C(\hat{F}_1(\zeta_1), 1) \leq 2 \left( \epsilon + L \cdot \frac{c_0}{B} \right).$$

Moreover, we have

$$C(\hat{F}_1(\zeta_1), 1) = \int_0^{F_1(\zeta_1)} \int_0^1 \underbrace{\frac{\partial^2}{\partial u \partial v} C(u, v)}_{\geq \ell} \, \mathrm{d}v \mathrm{d}u \geq \ell \hat{F}_1(\zeta_1),$$

and hence,

$$\hat{F}_1(\zeta_1) \le \frac{2}{\ell}\left(\epsilon + L \cdot \frac{c_0}{B}\right).$$

The similar argument gives

$$\hat{F}_2(\zeta_1) \le \frac{2}{\ell}\left(\epsilon + L \cdot \frac{c_0}{B}\right).$$

By combining these bounds, we obtain

$$C(\hat{F}_1(\zeta_1), \hat{F}_2(\zeta_1)) = \int_0^{\hat{F}_1(\zeta_1)} \int_0^{\hat{F}_2(\zeta_1)} \underbrace{\frac{\partial^2}{\partial u \partial v} C(u,v)}_{\le L} \, dv du$$

$$\le L \cdot \hat{F}_1(\zeta_1)\hat{F}_2(\zeta_1) \le \frac{4L}{\ell^2}\left(\epsilon + L \cdot \frac{c_0}{B}\right)^2. \tag{27}$$

Thus we get the bound on the term $C(\hat{F}_1(\zeta_1), \hat{F}_2(\zeta_1))$.

Moreover, we have

$$C(F_1(\zeta_1), F_2(\zeta_1)) = \int_0^{F_1(\zeta_1)} \int_0^{F_2(\zeta_1)} \underbrace{\frac{\partial^2}{\partial u \partial v} C(u,v)}_{\le L} \, dv du$$

$$\le L \cdot F_1(\zeta_1)F_2(\zeta_1) \le L \cdot \left(\frac{c_0}{B}\right)^2, \tag{28}$$

where we use Assumption 1-(1) for the last inequality.

Then, by taking the absolute value of the both sides of (26), we have

$$\left| C(\hat{F}_1(\zeta_1), 1) - C(F_1(\zeta_1), 1) \right|$$

$$= \left| \hat{r}_{1,1} - r_{1,1} + w_1 \cdot C(\hat{F}_1(\zeta_1), \hat{F}_2(\zeta_1)) - w_{1,1}^* \cdot C(F_1(\zeta_1), F_2(\zeta_1)) \right|$$

$$\le |\hat{r}_{1,1} - r_{1,1}| + \left| w_1 \cdot C(\hat{F}_1(\zeta_1), \hat{F}_2(\zeta_1)) - w_{1,1}^* \cdot C(F_1(\zeta_1), F_2(\zeta_1)) \right|$$

$$\le \epsilon + \max\left\{ C(\hat{F}_1(\zeta_1), \hat{F}_2(\zeta_1)), C(F_1(\zeta_1), F_2(\zeta_1)) \right\}$$

$$\le \epsilon + \max\left\{ \frac{4L}{\ell^2}\left(\epsilon + L \cdot \frac{c_0}{B}\right)^2, L \cdot \left(\frac{c_0}{B}\right)^2 \right\}$$

$$\le \epsilon + \frac{4L}{\ell^2}\left(\epsilon + L \cdot \frac{c_0}{B}\right)^2,$$

where we use the triangle inequality for the first inequality, $0 \le w_1, w_{1,1}^* \le 1$ for the second one, and (27), (28) for the third one. Since $\left| C(\hat{F}_1(\zeta_1), 1) - C(F_1(\zeta_1), 1) \right| \ge \ell \left| \hat{F}_1(\zeta_1) - F_1(\zeta_1) \right| = \ell |\Delta_{1,1}|$, we obtain

$$|\Delta_{1,1}| \le \frac{1}{\ell}\left[ \epsilon + \frac{4L}{\ell^2}\left(\epsilon + L \cdot \frac{c_0}{B}\right)^2 \right] \le \frac{1}{\ell}\left[ \epsilon + \frac{16L\delta^{-2}}{\ell^2}\left(\epsilon + L \cdot \frac{c_0}{B}\right)^2 \right],$$

where the last term coincides to the right hand side of (25) with $b = 1$.

We can obtain the bound for $k = 2$ by utilizing the same argument. Thus we get (25) for $b = 1$.

Assume that (25) holds for $b = b'$, i.e.,

$$|\Delta_{b',k}| \le \left[ \epsilon + \frac{16L\delta^{-2}}{\ell^2}\left(\epsilon + L \cdot \frac{c_0}{B}\right)^2 \right] \cdot \left[ \left(\frac{1}{\ell} + \frac{B}{4c_0 L}\right)\left(1 - \frac{8c_0 L}{B\ell\delta}\right)^{-b'+1} - \frac{B}{4c_0 L} \right]$$

holds for $k = 1, 2$. We consider the case $b = b' + 1$. This bound gives that by taking $\epsilon \leq \frac{c_\epsilon}{B}$ with a sufficiently small $c_\epsilon$,

$$
\begin{aligned}
|\Delta_{b',k}| &\leq \left[ \epsilon + \frac{16 L \delta^{-2}}{\ell^2} \left( \epsilon + L \cdot \frac{c_0}{B} \right)^2 \right] \cdot \left( \frac{1}{\ell} + \frac{B}{4 c_0 L} \right) \left( 1 - \frac{8 c_0 L}{B \ell \delta} \right)^{-b'+1} \\
&\leq \left[ \epsilon + \frac{16 L \delta^{-2}}{\ell^2} \left( \epsilon + L \cdot \frac{c_0}{B} \right)^2 \right] \cdot \left( \frac{1}{\ell} + \frac{B}{4 c_0 L} \right) \left( 1 - \frac{8 c_0 L}{B \ell \delta} \right)^{-B} \\
&\leq \left[ \epsilon + \frac{16 L \delta^{-2}}{\ell^2} \left( \epsilon + L \cdot \frac{c_0}{B} \right)^2 \right] \cdot \left( \frac{1}{\ell} + \frac{B}{4 c_0 L} \right) \exp\left( \frac{8 c_0 L}{\ell \delta} \right) \\
&\leq \left[ \epsilon + \frac{\ell \log B}{2 c_0^2} \left( \epsilon + L \cdot \frac{c_0}{B} \right)^2 \right] \cdot \left( \frac{1}{\ell} + \frac{B}{4 c_0 L} \right) B^\tau && (29) \\
&\lesssim \frac{1}{B^{1-\tau}}, && (30)
\end{aligned}
$$

where we use $1 - x \leq e^{-x}$ in the third inequality and the definition of $\delta$ in the fourth inequality. This and $F_k(\zeta_{b'}) < 1 - \delta$ gives $\hat{F}_k(\zeta_{b'}) < 1 - \frac{\delta}{2}$ for sufficiently large $B$.

We remind

$$
\hat{r}_{b'+1,1} = \hat{q}_{\{1\},b'} - w_1 \cdot \hat{q}_{\{1,2\},b'}, \tag{31}
$$

$$
r_{b'+1,1} = q_{\{1\},b'} - w^*_{b'+1,1} \cdot q_{\{1,2\},b'}, \tag{32}
$$

Now, we consider integral representation of $\hat{q}_{\{1\},b'}$ and $\hat{q}_{\{1,2\},b'}$ as

$$
\begin{aligned}
\hat{q}_{\{1\},b'} &= C(\hat{F}_1(\zeta_{b'+1}), 1) - C(\hat{F}_1(\zeta_{b'}), 1) - C((\zeta_{b'+1}), \hat{F}_2(\zeta_{b'})) + C(\hat{F}_1(\zeta_{b'}), \hat{F}_2(\zeta_{b'})) \\
&= \int_{\hat{F}_1(\zeta_{b'})}^{\hat{F}_1(\zeta_{b'+1})} \int_{\hat{F}_2(\zeta'_b)}^{1} \frac{\partial^2}{\partial u \partial v} C(u, v) \mathrm{d}v \mathrm{d}u, && (33) \\
\hat{q}_{\{1,2\},b'} &= C(\hat{F}_1(\zeta_{b'+1}), \hat{F}_2(\zeta_{b'+1})) - C(\hat{F}_1(\zeta_{b'+1}), \hat{F}_2(\zeta_{b'})) \\
&\quad - C(\hat{F}_1(\zeta_{b'}), \hat{F}_2(\zeta_{b'+1})) + C(\hat{F}_1(\zeta_{b'}), \hat{F}_2(\zeta_{b'})) \\
&= \int_{\hat{F}_1(\zeta_{b'})}^{\hat{F}_1(\zeta_{b'+1})} \int_{\hat{F}_2(\zeta'_b)}^{\hat{F}_2(\zeta_{b'+1})} \frac{\partial^2}{\partial u \partial v} C(u, v) \mathrm{d}v \mathrm{d}u.
\end{aligned}
$$

The same expression holds for $q_{\{1\},b'}$ and $q_{\{1,2\},b'}$ by replacing $\hat{F}_1$ and $\hat{F}_2$ with $F_1$ and $F_2$. By taking the difference between (31) and (32), we obtain

$$
\hat{r}_{b'+1,1} - r_{b'+1,1} = \hat{q}_{\{1\},b'} - \hat{q}_{\{1\},b'} - w_1 \cdot \hat{q}_{\{1,2\},b'} + w^*_{1,1} \cdot q_{\{1,2\},b'}. \tag{34}
$$

Similar to the case $b = 1$, we first evaluate the term $\hat{q}_{\{1,2\},b'}$ (note that $\hat{q}_{\{1,2\},b'} = C(\hat{F}_1(\zeta_1), \hat{F}_2(\zeta_1))$). By rearranging terms, we have

$$
\begin{aligned}
\hat{q}_{\{1\},b'} - w_1 \cdot \hat{q}_{\{1,2\},b'} &= \hat{r}_{b'+1,1} - r_{b'+1,1} + q_{\{1\},b'} - w^*_{1,1} \cdot q_{\{1,2\},b'} \\
&\leq \epsilon + q_{\{1\},b'} \\
&= \epsilon + \int_{F_1(\zeta_{b'})}^{F_1(\zeta_{b'+1})} \int_{F_2(\zeta'_b)}^{1} \underbrace{\frac{\partial^2}{\partial u \partial v} C(u, v)}_{\leq L} \mathrm{d}v \mathrm{d}u \\
&\leq \epsilon + L \cdot (F_1(\zeta_{b'+1}) - F_1(\zeta_{b'}))(1 - F_2(\zeta'_b)) \\
&\leq \epsilon + L \cdot \frac{c_0}{B}, && (35)
\end{aligned}
$$

where the first inequality follows from $\hat{r}_{b'+1,1} - r_{b'+1,1} \leq \epsilon$ by (14) and $q_{\{1,2\},b'} \geq 0$, and the last inequality, follows from Assumption 1-(1) and $1 - F_2(\zeta_b) < 1$. Moreover, the left hand side is lower

bounded by

$$\hat{q}_{\{1\},b'} - w_1 \cdot \hat{q}_{\{1,2\},b'} \geq (1 - w_1)\hat{q}_{\{1\},b'}$$

$$= \frac{1}{2} \int_{\hat{F}_1(\zeta_{b'})}^{\hat{F}_1(\zeta_{b'+1})} \int_{\hat{F}_2(\zeta_b')}^1 \underbrace{\frac{\partial^2}{\partial u \partial v} C(u, v)}_{\geq \ell} \, dv du$$

$$\geq \frac{1}{2} \cdot \ell \Big( \hat{F}_1(\zeta_{b'+1}) - \hat{F}_1(\zeta_{b'}) \Big)(1 - \hat{F}_2(\zeta_{b'})),$$

where we use $1 - w_1 = \frac{1}{2}$ and (33) for the equality. Combining this with (35), we have

$$\hat{F}_1(\zeta_{b'+1}) - \hat{F}_1(\zeta_{b'}) \leq \frac{2}{\ell(1 - \hat{F}_2(\zeta_{b'}))} \Big( \epsilon + L \cdot \frac{c_0}{B} \Big). \tag{36}$$

This and the triangle inequality give

$$|\Delta_{b'+1,1}| = \left| \Big( \hat{F}_1(\zeta_{b'+1}) - \hat{F}_1(\zeta_{b'}) \Big) + \Big( \hat{F}_1(\zeta_{b'}) - F_1(\zeta_{b'}) \Big) + (F_1(\zeta_{b'}) - F_1(\zeta_{b'+1})) \right|$$

$$\leq \left| \hat{F}_1(\zeta_{b'+1}) - \hat{F}_1(\zeta_{b'}) \right| + \left| \hat{F}_1(\zeta_{b'}) - F_1(\zeta_{b'}) \right| + |F_1(\zeta_{b'}) - F_1(\zeta_{b'+1})|$$

$$\leq \frac{2}{\ell(1 - \hat{F}_2(\zeta_{b'}))} \Big( \epsilon + L \cdot \frac{c_0}{B} \Big) + |\Delta_{b',1}| + \frac{c_0}{B},$$

which we will use in the latter of this proof.

The same bound as (36) holds for $k = 2$, which gives

$$\Big( \hat{F}_1(\zeta_{b'+1}) - \hat{F}_1(\zeta_{b'}) \Big) \Big( \hat{F}_2(\zeta_{b'+1}) - \hat{F}_2(\zeta_{b'}) \Big) \leq \frac{4}{\ell^2 \Big( 1 - \hat{F}_1(\zeta_{b'}) \Big) \Big( 1 - \hat{F}_2(\zeta_{b'}) \Big)} \Big( \epsilon + L \cdot \frac{c_0}{B} \Big)^2.$$

Thus, we obtain

$$\hat{q}_{\{1,2\},b'} = \int_{\hat{F}_1(\zeta_{b'})}^{\hat{F}_1(\zeta_{b'+1})} \int_{\hat{F}_2(\zeta_b')}^{\hat{F}_2(\zeta_{b'+1})} \underbrace{\frac{\partial^2}{\partial u \partial v} C(u, v)}_{\leq L} \, dv du \tag{37}$$

$$\leq L \cdot \Big( \hat{F}_1(\zeta_{b'+1}) - \hat{F}_1(\zeta_{b'}) \Big) \Big( \hat{F}_2(\zeta_{b'+1}) - \hat{F}_2(\zeta_{b'}) \Big)$$

$$\leq \frac{4L}{\ell^2 \Big( 1 - \hat{F}_1(\zeta_{b'}) \Big) \Big( 1 - \hat{F}_2(\zeta_{b'}) \Big)} \Big( \epsilon + L \cdot \frac{c_0}{B} \Big)^2.$$

Moreover, we have

$$q_{\{1,2\},b'|x} = \int_{F_1(\zeta_{b'})}^{F_1(\zeta_{b'+1})} \int_{F_2(\zeta_b')}^{F_2(\zeta_{b'+1})} \underbrace{\frac{\partial^2}{\partial u \partial v} C(u, v)}_{\leq L} \, dv du \tag{38}$$

$$\leq L \cdot (F_1(\zeta_{b'+1}) - F_1(\zeta_{b'}))(F_2(\zeta_{b'+1}) - F_2(\zeta_{b'}))$$

$$\leq L \cdot \Big( \frac{c_0}{B} \Big)^2,$$

where we use Assumption 1-(1) for the last inequality.

Then, by taking the absolute value of the both sides of (34), we have

$$
\left|\hat{q}_{\{1\},b'|x} - q_{\{1\},b'|x}\right|
$$
$$
= \left|\hat{r}_{b'+1,1} - r_{b'+1,1} + w_1 \cdot \hat{q}_{\{1,2\},b'|x} - w^*_{b'+1,1} \cdot q_{\{1,2\},b'|x}\right|
$$
$$
\leq \left|\hat{r}_{b'+1,1} - r_{b'+1,1}\right| + \left|w_1 \cdot \hat{q}_{\{1,2\},b'|x} - w^*_{b'+1,1} \cdot q_{\{1,2\},b'|x}\right|
$$
$$
\leq \epsilon + \max\left\{\hat{q}_{\{1,2\},b'|x}, q_{\{1,2\},b'|x}\right\}
$$
$$
\leq \epsilon + \max\left\{\frac{4L}{\ell^2\left(1 - \hat{F}_1(\zeta_{b'})\right)\left(1 - \hat{F}_2(\zeta_{b'})\right)}\left(\epsilon + L \cdot \frac{c_0}{B}\right)^2, L \cdot \left(\frac{c_0}{B}\right)^2\right\}
$$
$$
\leq \epsilon + \frac{4L}{\ell^2\left(1 - \hat{F}_1(\zeta_{b'})\right)\left(1 - \hat{F}_2(\zeta_{b'})\right)}\left(\epsilon + L \cdot \frac{c_0}{B}\right)^2, \tag{39}
$$

where we use the triangle inequality for the first inequality, $0 \leq w_1, w^*_{b'+1,1} \leq 1$ for the second one, and (37) and (38) for the third one.

Then, we evaluate the left hand side. We have

$$
\left|\hat{q}_{\{1\},b'} - q_{\{1\},b'}\right|
$$
$$
= \left|\int_{\hat{F}_1(\zeta_{b'})}^{\hat{F}_1(\zeta_{b'+1})} \int_{\hat{F}_2(\zeta'_b)}^1 \frac{\partial^2}{\partial u \partial v}C(u,v)\mathrm{d}v\mathrm{d}u - \int_{F_1(\zeta_{b'})}^{F_1(\zeta_{b'+1})} \int_{F_2(\zeta'_b)}^1 \frac{\partial^2}{\partial u \partial v}C(u,v)\mathrm{d}v\mathrm{d}u\right|
$$

and

$$
\int_{F_1(\zeta_{b'})}^{F_1(\zeta_{b'+1})} \int_{F_2(\zeta'_b)}^1 \frac{\partial^2}{\partial u \partial v}C(u,v)\mathrm{d}v\mathrm{d}u - \int_{\hat{F}_1(\zeta_{b'})}^{\hat{F}_1(\zeta_{b'+1})} \int_{\hat{F}_2(\zeta'_b)}^1 \frac{\partial^2}{\partial u \partial v}C(u,v)\mathrm{d}v\mathrm{d}u
$$
$$
= \int_{F_1(\zeta_{b'})}^{F_1(\zeta_{b'+1})} \frac{\partial}{\partial u}C(u,1)\mathrm{d}u - \int_{\hat{F}_1(\zeta_{b'})}^{\hat{F}_1(\zeta_{b'+1})} \frac{\partial}{\partial u}C(u,1)\mathrm{d}u
$$
$$
- \int_{F_1(\zeta_{b'})}^{F_1(\zeta_{b'+1})} \frac{\partial}{\partial u}C(u, F_2(\zeta_{b'}))\mathrm{d}u + \int_{\hat{F}_1(\zeta_{b'})}^{\hat{F}_1(\zeta_{b'+1})} \frac{\partial}{\partial u}C(u, \hat{F}_2(\zeta_{b'}))\mathrm{d}u
$$
$$
= \int_{\hat{F}_1(\zeta_{b'+1})}^{F_1(\zeta_{b'+1})} \frac{\partial}{\partial u}C(u,1)\mathrm{d}u - \int_{\hat{F}_1(\zeta_{b'})}^{F_1(\zeta_{b'})} \frac{\partial}{\partial u}C(u,1)\mathrm{d}u
$$
$$
- \int_{\hat{F}_1(\zeta_{b'+1})}^{\hat{F}_1(\zeta_{b'})} \frac{\partial}{\partial u}C(u, F_2(\zeta_{b'}))\mathrm{d}u + \int_{F_1(\zeta_{b'+1})}^{F_1(\zeta_{b'})} \frac{\partial}{\partial u}C(u, F_2(\zeta_{b'}))\mathrm{d}u
$$
$$
+ \int_{\hat{F}_1(\zeta_{b'})}^{\hat{F}_1(\zeta_{b'+1})} \left[\frac{\partial}{\partial u}C(u, \hat{F}_2(\zeta_{b'})) - \frac{\partial}{\partial u}C(u, F_2(\zeta_{b'}))\right]\mathrm{d}u
$$
$$
= \int_{\hat{F}_1(\zeta_{b'+1})}^{F_1(\zeta_{b'+1})} \frac{\partial}{\partial u}C(u,1)\mathrm{d}u - \int_{\hat{F}_1(\zeta_{b'})}^{F_1(\zeta_{b'})} \frac{\partial}{\partial u}C(u,1)\mathrm{d}u
$$
$$
- \int_{\hat{F}_1(\zeta_{b'+1})}^{F_1(\zeta_{b'+1})} \frac{\partial}{\partial u}C(u, F_2(\zeta_{b'}))\mathrm{d}u + \int_{\hat{F}_1(\zeta_{b'})}^{F_1(\zeta_{b'})} \frac{\partial}{\partial u}C(u, F_2(\zeta_{b'}))\mathrm{d}u \tag{40}
$$
$$
+ \int_{\hat{F}_1(\zeta_{b'})}^{\hat{F}_1(\zeta_{b'+1})} \left[\frac{\partial}{\partial u}C(u, \hat{F}_2(\zeta_{b'})) - \frac{\partial}{\partial u}C(u, F_2(\zeta_{b'}))\right]\mathrm{d}u.
$$

By the mean value theorem for integrals, there exist constants

$$
u_{b',1}, u_{b',2} \in \left[\min\left\{\hat{F}_1(\zeta_{b'}), F_1(\zeta_{b'})\right\}, \max\left\{\hat{F}_1(\zeta_{b'}), F_1(\zeta_{b'})\right\}\right]
$$

and

$$
u_{b'+1,1}, u_{b'+1,2} \in \left[\min\left\{\hat{F}_1(\zeta_{b'+1}), F_1(\zeta_{b'+1})\right\}, \max\left\{\hat{F}_1(\zeta_{b'+1}), F_1(\zeta_{b'+1})\right\}\right]
$$

such that

$$(40) = -\Delta_{b'+1,1} \cdot \frac{\partial}{\partial u} C(u_{b'+1,1}, 1) + \Delta_{b',1} \frac{\partial}{\partial u} C(u_{b',1}, 1)$$

$$+ \Delta_{b'+1,1} \cdot \frac{\partial}{\partial u} C(u_{b'+1,2}, F_2(\zeta_{b'+1})) - \Delta_{b'+1,1} \cdot \frac{\partial}{\partial u} C(u_{b',2}, F_2(\zeta_{b'}))$$

$$+ \int_{\hat{F}_1(\zeta_{b'})}^{\hat{F}_1(\zeta_{b'+1})} \left[ \frac{\partial}{\partial u} C(u, \hat{F}_2(\zeta_{b'})) - \frac{\partial}{\partial u} C(u, F_2(\zeta_{b'})) \right] \mathrm{d}u.$$

We denote

$$(\mathrm{I}) = \Delta_{b'+1,1} \cdot \frac{\partial}{\partial u} C(u_{b'+1,1}, 1) - \Delta_{b',1} \frac{\partial}{\partial u} C(u_{b',1}, 1)$$

$$- \Delta_{b'+1,1} \cdot \frac{\partial}{\partial u} C(u_{b'+1,2}, F_2(\zeta_{b'+1})) + \Delta_{b'+1,1} \cdot \frac{\partial}{\partial u} C(u_{b',2}, F_2(\zeta_{b'})),$$

$$(\mathrm{II}) = \int_{\hat{F}_1(\zeta_{b'})}^{\hat{F}_1(\zeta_{b'+1})} \left[ \frac{\partial}{\partial u} C(u, \hat{F}_2(\zeta_{b'})) - \frac{\partial}{\partial u} C(u, F_2(\zeta_{b'})) \right] \mathrm{d}u.$$

We evaluate the each term. First, we have

$$(\mathrm{I}) = (\Delta_{b'+1,1} - \Delta_{b',1}) \frac{\partial}{\partial u} C(u_{b',1}, 1) + \Delta_{b'+1,1} \left[ \frac{\partial}{\partial u} C(u_{b'+1,1}, 1) - \frac{\partial}{\partial u} C(u_{b',1}, 1) \right]$$

$$- (\Delta_{b'+1,1} - \Delta_{b',1}) \frac{\partial}{\partial u} C(u_{b',2}, F_2(\zeta_{b'}))$$

$$- \Delta_{b'+1,1} \left[ \frac{\partial}{\partial u} C(u_{b'+1,2}, F_2(\zeta_{b'+1})) - \frac{\partial}{\partial u} C(u_{b',2}, F_2(\zeta_{b'})) \right]$$

$$= (\Delta_{b'+1,1} - \Delta_{b',1}) \left[ \frac{\partial}{\partial u} C(u_{b',1}, 1) - \frac{\partial}{\partial u} C(u_{b',2}, F_2(\zeta_{b'})) \right]$$

$$+ \Delta_{b'+1,1} \left[ \frac{\partial}{\partial u} C(u_{b'+1,1}, 1) - \frac{\partial}{\partial u} C(u_{b',1}, 1) \right]$$

$$+ \Delta_{b'+1,1} \left[ \frac{\partial}{\partial u} C(u_{b'+1,2}, F_2(\zeta_{b'+1})) - \frac{\partial}{\partial u} C(u_{b',2}, F_2(\zeta_{b'})) \right].$$

Thus, we obtain

$$|(\mathrm{I})| \geq \left| (\Delta_{b'+1,1} - \Delta_{b',1}) \left[ \frac{\partial}{\partial u} C(u_{b',1}, 1) - \frac{\partial}{\partial u} C(u_{b',2}, F_2(\zeta_{b'})) \right] \right|$$

$$- \left| \Delta_{b'+1,1} \left[ \frac{\partial}{\partial u} C(u_{b'+1,1}, 1) - \frac{\partial}{\partial u} C(u_{b',1}, 1) \right] \right|$$

$$- \left| \Delta_{b'+1,1} \left[ \frac{\partial}{\partial u} C(u_{b'+1,2}, F_2(\zeta_{b'+1})) - \frac{\partial}{\partial u} C(u_{b',2}, F_2(\zeta_{b'})) \right] \right|,$$

where we use the triangle inequality.

Moreover, we can bound the terms in the above inequality by

$$\left| \frac{\partial}{\partial u} C(u_{b'+1,1}, 1) - \frac{\partial}{\partial u} C(u_{b',1}, 1) \right| = \left| \int_{u_{b',1}}^{u_{b'+1,1}} \underbrace{\frac{\partial^2}{\partial u^2} C(u, 1)}_{\leq L} \mathrm{d}u \right| \leq L(u_{b'+1,1} - u_{b',1})$$

$$\leq L \left( |\Delta_{b'+1,1}| + \frac{c_0}{B} \right),$$

and

$$
\left| \frac{\partial}{\partial u} C(u_{b'+1,2}, F_2(\zeta_{b'+1})) - \frac{\partial}{\partial u} C(u_{b',2}, F_2(\zeta_{b'})) \right|
$$

$$
\leq \left| \frac{\partial}{\partial u} C(u_{b'+1,2}, F_2(\zeta_{b'+1})) - \frac{\partial}{\partial u} C(u_{b'+1,2}, F_2(\zeta_{b'})) \right|
$$

$$
+ \left| \frac{\partial}{\partial u} C(u_{b'+1,2}, F_2(\zeta_{b'})) - \frac{\partial}{\partial u} C(u_{b',2}, F_2(\zeta_{b'})) \right|
$$

$$
= \left| \int_{F_2(\zeta_{b'})}^{F_2(\zeta_{b'+1})} \underbrace{\frac{\partial^2}{\partial v^2} C(u_{b'+1,2}, v)}_{\leq L} \, \mathrm{d}v \right| + \left| \int_{u_{b'+1,1}}^{u_{b',2}} \underbrace{\frac{\partial^2}{\partial u^2} C(u, F_2(\zeta_{b'}))}_{\leq L} \, \mathrm{d}u \right|
$$

$$
\leq L \cdot \frac{c_0}{B} + L \left( \frac{c_0}{B} + |\Delta_{b'+1,1}| \right)
$$

$$
= L \left( |\Delta_{b'+1,1}| + \frac{2c_0}{B} \right),
$$

where we use Assumption 1-(1) for the first term and $|u_{b',2} - u_{b'+1,2}| \leq \Delta_{b'+1,1} + \frac{c_0}{B}$ for the second term. Moreover, we have

$$
|(\mathrm{II})| = \left| \int_{\hat{F}_1(\zeta_{b'})}^{\hat{F}_1(\zeta_{b'+1})} \int_{F_2(\zeta_{b'})}^{\hat{F}_2(\zeta_{b'})} \underbrace{\frac{\partial^2}{\partial u \partial v} C(u, v)}_{\leq L} \, \mathrm{d}v \mathrm{d}u \right| \leq L \frac{2}{\ell \delta} \left( \epsilon + L \cdot \frac{c_0}{B} \right) |\Delta_{b',2}| \leq \frac{4 c_0 L^2}{B \ell \delta} |\Delta_{b',2}|,
$$

where we use $\epsilon \leq \frac{c_0}{B}$ in the last inequality.

Then, (39) gives

$$
|\Delta_{b'+1,1} - \Delta_{b',1}| \cdot \left| \frac{\partial}{\partial u} C(u_{b',1}, 1) - \frac{\partial}{\partial u} C(u_{b',2}, F_2(\zeta_{b'})) \right|
$$

$$
\leq \epsilon + \frac{4L}{\ell^2 \left( 1 - \hat{F}_1(\zeta_{b'}) \right) \left( 1 - \hat{F}_2(\zeta_{b'}) \right)} \left( \epsilon + L \cdot \frac{c_0}{B} \right)^2
$$

$$
+ \frac{4 c_0 L^2}{B \ell \delta} |\Delta_{b',2}| + L |\Delta_{b'+1,1}| \left( |\Delta_{b'+1,1}| + \frac{3 c_0}{B} \right).
$$

Moreover, by the triangle inequality, for a sufficiently large $B$ satisfying $L \cdot \frac{c_0}{B} \leq \frac{\ell \delta}{2}$, we have

$$
\left| \frac{\partial}{\partial u} C(u_{b',1}, 1) - \frac{\partial}{\partial u} C(u_{b',2}, F_2(\zeta_{b'})) \right|
$$

$$
\geq \left| \frac{\partial}{\partial u} C(u_{b',2}, 1) - \frac{\partial}{\partial u} C(u_{b',2}, F_2(\zeta_{b'})) \right| - \left| \frac{\partial}{\partial u} C(u_{b',1}, 1) - \frac{\partial}{\partial u} C(u_{b',2}, 1) \right|
$$

$$
= \left| \int_{F_2(\zeta_{b'})}^{1} \underbrace{\frac{\partial^2}{\partial v^2} C(u_{b',2}, v)}_{\geq \ell} \, \mathrm{d}v \right| - \left| \int_{u_{b',1}}^{u_{b',2}} \underbrace{\frac{\partial^2}{\partial u^2} C(u, 1))}_{\leq L} \, \mathrm{d}u \right|
$$

$$
\geq \ell (1 - F_2(\zeta_{b'})) - L |u_{b',2} - u_{b',1}|
$$

$$
\geq \ell \delta - L \cdot \frac{c_0}{B} \geq \frac{\ell \delta}{2},
$$

where the third inequality follows from $F_2(\zeta_b) \leq 1 - \delta$ and

$$
|u_{b',2} - u_{b',1}| \leq \max \left\{ \hat{F}_1(\zeta_{b'}), F_1(\zeta_{b'}) \right\} - \min \left\{ \hat{F}_1(\zeta_{b'}), F_1(\zeta_{b'}) \right\} = |\Delta_{b',k}| \leq \frac{c_0}{B}
$$

by (30).

Then, we have

$$|\Delta_{b'+1,1} - \Delta_{b',1}|$$

$$\leq \frac{2}{\ell\delta}\left[\epsilon + \frac{4L}{\ell^2\left(1 - \hat{F}_1(\zeta_{b'})\right)\left(1 - \hat{F}_2(\zeta_{b'})\right)}\left(\epsilon + L\cdot\frac{c_0}{B}\right)^2 + \frac{4c_0 L^2}{B\ell\delta}|\Delta_{b',2}| + L|\Delta_{b'+1,1}|\left(|\Delta_{b'+1,1}| + \frac{3c_0}{B}\right)\right]$$

$$\leq \frac{2}{\ell\delta}\left[\epsilon + \frac{4L}{\ell^2\left(1 - \hat{F}_1(\zeta_{b'})\right)\left(1 - \hat{F}_2(\zeta_{b'})\right)}\left(\epsilon + L\cdot\frac{c_0}{B}\right)^2 + \frac{4c_0 L^2}{B\ell\delta}|\Delta_{b',2}|\right] + \frac{2L}{\ell\delta}|\Delta_{b'+1,1}|\left(|\Delta_{b'+1,1}| + \frac{3c_0}{B}\right).$$

We consider two cases (i) $\Delta_{b'+1,1} \leq \frac{c_0}{B}$ and (ii) $\Delta_{b'+1,1} > \frac{c_0}{B}$. If (i) holds, we have

$$|\Delta_{b'+1,1} - \Delta_{b',1}| \leq \frac{2}{\ell\delta}\left[\epsilon + \frac{4L}{\ell^2\left(1 - \hat{F}_1(\zeta_{b'})\right)\left(1 - \hat{F}_2(\zeta_{b'})\right)}\left(\epsilon + L\cdot\frac{c_0}{B}\right)^2 + \frac{4c_0 L^2}{B\ell\delta}|\Delta_{b',2}|\right] + \frac{8c_0 L}{B\ell\delta}|\Delta_{b'+1,1}|.$$

By using the triangle inequality again, we obtain

$$|\Delta_{b'+1,1}| \leq |\Delta_{b',1}| + \frac{2}{\ell\delta}\left[\epsilon + \frac{4L}{\ell^2\left(1 - \hat{F}_1(\zeta_{b'})\right)\left(1 - \hat{F}_2(\zeta_{b'})\right)}\left(\epsilon + L\cdot\frac{c_0}{B}\right)^2 + \frac{4c_0 L^2}{B\ell\delta}|\Delta_{b',2}|\right] + \frac{8c_0 L}{B\ell\delta}|\Delta_{b'+1,1}|.$$

Finally, by rearranging the above inequality and using the induction hypothesis, we have

$$\left(1 - \frac{8c_0 L}{B\ell\delta}\right)|\Delta_{b'+1,1}| \leq \left[\epsilon + \frac{16L\delta^{-2}}{\ell^2}\left(\epsilon + L\cdot\frac{c_0}{B}\right)^2\right]\cdot\left[\left(\frac{1}{\ell} + \frac{B}{4c_0 L}\right)\left(1 - \frac{8c_0 L}{B\ell\delta}\right)^{-b'+1} - \frac{B}{4c_0 L}\right]$$

$$+ \frac{2}{\ell\delta}\left[\epsilon + \frac{4L}{\ell^2\left(1 - \hat{F}_1(\zeta_{b'})\right)\left(1 - \hat{F}_2(\zeta_{b'})\right)}\left(\epsilon + L\cdot\frac{c_0}{B}\right)^2\right]$$

$$\leq \left[\epsilon + \frac{16L\delta^{-2}}{\ell^2}\left(\epsilon + L\cdot\frac{c_0}{B}\right)^2\right]\cdot\left[\left(\frac{1}{\ell} + \frac{B}{4c_0 L}\right)\left(1 - \frac{8c_0 L}{B\ell\delta}\right)^{-b'+1} - \frac{B}{4c_0 L}\right]$$

$$+ \frac{2}{\ell\delta}\left[\epsilon + \frac{16L\delta^{-2}}{\ell^2}\left(\epsilon + L\cdot\frac{c_0}{B}\right)^2\right],$$

where we use $F_k < 1 - \frac{\delta}{2}$ for the last inequality, and hence,

$$|\Delta_{b'+1,1}| \leq \left[\epsilon + \frac{16L\delta^{-2}}{\ell^2}\left(\epsilon + L\cdot\frac{c_0}{B}\right)^2\right]\cdot\left[\left(\frac{1}{\ell} + \frac{B}{4c_0 L}\right)\left(1 - \frac{8c_0 L}{B\ell\delta}\right)^{-b'+1} - \frac{B}{4c_0 L}\right]\left(1 - \frac{8c_0 L}{B\ell\delta}\right)^{-1}$$

$$+ \left(1 - \frac{8c_0 L}{B\ell\delta}\right)^{-1}\cdot\frac{2}{\ell\delta}\left[\epsilon + \frac{16L\delta^{-2}}{\ell^2}\left(\epsilon + L\cdot\frac{c_0}{B}\right)^2\right]$$

$$= \left[\epsilon + \frac{16L\delta^{-2}}{\ell^2}\left(\epsilon + L\cdot\frac{c_0}{B}\right)^2\right]\cdot\left[\left(\frac{1}{\ell} + \frac{B}{4c_0 L}\right)\left(1 - \frac{8c_0 L}{B\ell\delta}\right)^{-(b'+1)+1} - \frac{B}{4c_0 L}\right],$$

which ensures (25) for $b = b' + 1$ and $k = 1$. Since the same argument holds for $k = 2$, we obtain the conclusion for the case (i).

If (ii) holds, we have

$$|\Delta_{b'+1,1} - \Delta_{b',1}|$$

$$\leq \frac{2}{\ell\delta}\left[\epsilon + \frac{4L}{\ell^2\left(1 - \hat{F}_1(\zeta_{b'})\right)\left(1 - \hat{F}_2(\zeta_{b'})\right)}\left(\epsilon + L \cdot \frac{c_0}{B}\right)^2 + \frac{4c_0L^2}{B\ell\delta}|\Delta_{b',2}|\right] + \frac{8L}{\ell\delta}|\Delta_{b'+1,1}|^2.$$

Then, by using (30), we have $|\Delta_{b'+1,1}|^2 \lesssim B^{-2(1-\tau)}$. This and triangle inequality gives

$$|\Delta_{b'+1,1}| \leq |\Delta_{b',1}|$$

$$+ \frac{2}{\ell\delta}\left[\epsilon + \frac{4L}{\ell^2\left(1 - \hat{F}_1(\zeta_{b'})\right)\left(1 - \hat{F}_2(\zeta_{b'})\right)}\left(\epsilon + L \cdot \frac{c_0}{B}\right)^2 + \frac{3c_0L^2}{B\ell\delta}|\Delta_{b',2}|\right] + \frac{8L}{\ell\delta}|\Delta_{b'+1,1}|^2$$

$$\leq |\Delta_{b',1}| + \left(1 - \frac{8c_0L}{B\ell\delta}\right)^{-1}\frac{2}{\ell\delta}\left[\epsilon + \frac{16L}{\ell^2\left(1 - \hat{F}_1(\zeta_{b'})\right)\left(1 - \hat{F}_2(\zeta_{b'})\right)}\left(\epsilon + L \cdot \frac{c_0}{B}\right)^2 + \frac{4c_0L^2}{B\ell\delta}|\Delta_{b',2}|\right]$$

with taking a sufficiently large $B$. This gives the conclusion for the case (ii). $\qquad\square$

To obtain the bound with respect to $W_1$, we utilize the following lemma:

**Lemma 1** (Vallender (1974)). *Let $\mu$ and $\nu$ be cumulative distribution functions on $\mathbb{R}$ whose CDFs are defined by $F$ and $G$. Then,*

$$\int_{-\infty}^{+\infty} |F(t) - G(t)|\mathrm{d}t = W_1(\mu, \nu).$$

Then, we move to the proof of Theorem 2.

*Proof of Theorem 2.* Set $\delta_0$ in Assumption 1-(3) by $\delta_0 := \frac{8c_0L}{\ell}$. By using Lemma 3, we obtain

$$\begin{aligned}
W_1(\hat{F}_k, F_k) &= \int_{-\infty}^{+\infty} \left|\hat{F}_k(t) - F_k(t)\right|\mathrm{d}t \\
&= \int_0^T \left|\hat{F}_k(t) - F_k(t)\right|\mathrm{d}t \\
&= \underbrace{\sum_{b=1}^{b_0}\int_{\zeta_{b-1}}^{\zeta_b}\left|\hat{F}_k(t) - F_k(t)\right|\mathrm{d}t}_{\text{(I)}} + \underbrace{\int_{\zeta_{b_0}}^{\zeta_B}\left|\hat{F}_k(t) - F_k(t)\right|\mathrm{d}t}_{\text{(II)}}, \qquad (41)
\end{aligned}$$

where $b_0$ is defined by Assumption 1-(3). Then, we bound the each term.

We denote $\Delta_{b,k} := \hat{F}_k(\zeta_b) - F_k(\zeta_b)$ again. By using the bound (29), we have

$$|\Delta_{b,k}| \lesssim \left(\epsilon + \frac{\log B}{B^2}\right)B^{1+\tau}$$

Then, the first term can be bounded by

$$
\begin{aligned}
\text{(I)} &\leq \sum_{b=1}^{b_0} \int_{\zeta_{b-1}}^{\zeta_b} \max\left\{ \left| \hat{F}_k(t) - F_k(\zeta_{b-1}) \right|, \left| \hat{F}_k(t) - F_k(\zeta_b) \right| \right\} \mathrm{d}t \\
&\leq \sum_{b=1}^{b_0} \int_{\zeta_{b-1}}^{\zeta_b} \max\left\{ |\Delta_{b,k}|, \frac{c_0}{B} \right\} \mathrm{d}t \\
&\leq \sum_{b=1}^{b_0} \int_{\zeta_{b-1}}^{\zeta_b} \left( |\Delta_{b,k}| + \frac{c_0}{B} \right) \mathrm{d}t \\
&\lesssim \sum_{b=1}^{b_0} \left( \epsilon + \frac{\log B}{B^2} \right) B^{1+\tau} \cdot \frac{\zeta_B}{B} \\
&\leq \sum_{b=1}^{B} \left( \epsilon + \frac{\log B}{B^2} \right) B^{1+\tau} \cdot \frac{\zeta_B}{B} \\
&= \left( \epsilon + \frac{\log B}{B^2} \right) B^{1+\tau} \zeta_B.
\end{aligned}
\tag{42}
$$

For the second term, since $1 - F_k(\zeta_{b_0'}) \gtrsim \log^{-1} B$ by Assumption 1-(3), we have $\left| \hat{F}_k(t) - F_k(t) \right| \lesssim \log^{-1} B$ for $t \geq \zeta_{b_0}$ with sufficiently large $B$. This implies

$$
\text{(II)} \lesssim \int_{\zeta_{b_0}}^{\zeta_B} \log^{-1} B \, \mathrm{d}t \leq \frac{B - b_0}{B \log B} \zeta_B.
\tag{43}
$$

By substituting the bounds (42) and (43) into (41), we obtain the conclusion. $\qquad\square$

# E    IMPOSSIBILITY THEOREM

A limitation of the two-step algorithm is its presupposition that the copula $C$ is predetermined. Nevertheless, it has been established that the marginal distribution remains unidentifiable when this assumption is removed, which is known as Tsiatis's impossibility theorem [1975].

**Explanation of Tsiatis's Impossibility Theorem.**    We briefly elucidate this impossibility theorem. Assume the existence of a dataset $\mathcal{D} = \{(x^{(i)}, t^{(i)}, \delta^{(i)})\}_{i=1}^N$ derived from a joint probability distribution $\mathbb{P}$, represented as

$$
\Pr(t_1 \leq T_1, t_2 \leq T_2, \ldots, t_K \leq T_K) = C(F_1(t_1), F_2(t_2), \ldots, F_K(t_K))
$$

for a copula $C$ that deviates from the independence copula $C_{\mathrm{ind}}$ (i.e., $C \neq C_{\mathrm{ind}}$). Tsiatis's impossibility theorem suggests the existence of an alternative joint distribution $\mathbb{Q}$, represented as

$$
\Pr(t_1 \leq T_1', t_2 \leq T_2', \ldots, t_K \leq T_K') = C_{\mathrm{ind}}(G_1(t_1), G_2(t_2), \ldots, G_K(t_K)),
$$

where $G(t_k)$ is a CDF of a random variable $T_k'$, distinct from $T_k$. However, the dataset $\mathcal{D}_{\mathrm{ind}} = \{(x^{(i)}, t^{(i)}, \delta^{(i)})\}_{i=1}^N$ derived from $\mathbb{Q}$ is indistinguishable from $\mathcal{D}$. Consequently, no algorithm can differentiate the marginal distributions $F_k(t)$ and $G_k(t)$ from the dataset $\mathcal{D}(\approx \mathcal{D}_{\mathrm{ind}})$. This ambiguity does not arise if all realizations $(t_1, t_2, \ldots, t_K) \sim (T_1, T_2, \ldots, T_K)$ can be observed, as per Sklar's theorem (Theorem 1). Tsiatis's impossibility theorem holds because only the minimum of $\{t_1, t_2, \ldots, t_K\}$ can be observed in survival analysis.

The two-step algorithm we propose aligns with Tsiatis's impossibility theorem. The initial step's output $\hat{r}_{b,k}$ remains consistent for both $\mathbb{P}$ and $\mathbb{Q}$ if the indistinguishable dataset $\mathcal{D}$ or $\mathcal{D}_{\mathrm{ind}}$ is provided as input. The subsequent step of our algorithm yields $\hat{F}_k(t)$ if the true copula $C$ is given as input, while it produces $\hat{G}_k(t)$ if the independence copula $C_{\mathrm{ind}}$ is provided.

**Copula identifiability.** Owing to Tsiatis's impossibility theorem, it is necessary to make certain assumptions to identify the marginal distribution. In this context, Heckman & Honoré (1989); Deresa & Keilegom (2024) examine copula identifiability under the proportional hazard assumption. Furthermore, Czado & Keilegom (2022); Zhang et al. (2024) discuss the identifiability of copulas for other restricted classes of marginal distributions $F_k(t|x)$. Under the stronger assumption that the marginal distribution $F_k(t|x)$ is completely known, Schwarz et al. (2013) discuss the identifiability of Archimedean copulas and the unidentifiability of symmetric copulas.

## F ADDITIONAL RESULTS ON UPPER AND LOWER BOUNDS

### F.1 BOUNDS BASED ON THE CUMULATIVE INCIDENCE FUNCTION

We provide alternative bounds by utilizing $V_k(\zeta_b|x)$ (as defined in (3)):

$$\Pr(T \leq \zeta_b, \delta = k|x) \leq F_k(\zeta_b|x) \leq \Pr(T \leq \zeta_b|x)$$

$$\Leftrightarrow \qquad V_k(\zeta_b|x) \leq F_k(\zeta_b|x) \leq \sum_k V_k(\zeta_b|x).$$

By averaging over $x \sim X$, we can derive the same upper and lower bounds as in (Peterson, 1976):

$$\mathop{\mathbb{E}}_{x \sim X}[V_k(\zeta_b|x)] \leq F_k(\zeta_b) \leq \mathop{\mathbb{E}}_{x \sim X}\left[\sum_k V_k(\zeta_b|x)\right]. \tag{44}$$

The empirical bounds of these upper and lower bounds in (44) can be computed as:

$$\mathop{\mathbb{E}}_{x \sim X}[V_k(\zeta_b|x)] \approx \frac{1}{N} \sum_{(x^{(i)}, t^{(i)}, \delta^{(i)}) \in \mathcal{D}} \mathbb{1}_{t^{(i)} \leq \zeta_b, \delta^{(i)} = k}$$

$$\mathop{\mathbb{E}}_{x \sim X}\left[\sum_k V_k(\zeta_b|x)\right] \approx \frac{1}{N} \sum_{(x^{(i)}, t^{(i)}, \delta^{(i)}) \in \mathcal{D}} \mathbb{1}_{t^{(i)} \leq \zeta_b}.$$

These values are equivalent to empirical CDFs. However, they may not correspond to the actual bounds if the number of data points $N$ is insufficient. In such cases, the confidence intervals of these empirical CDFs should also be computed using methods like Greenwood's method [1926].

### F.2 EXAMPLES

We illustrate survival functions along with bounds for the flchain and support2 datasets in Fig. 6. The two graphs on the left depict average survival functions, while the four graphs in the center and right display individual survival functions.

For the average survival functions, we employed the Kaplan-Meier (KM) estimator (1958) and the copula-graphic (CG) estimator (Zheng & Klein, 1995) using the Frank copula (2) with parameters $\theta = -5$ and $\theta = 5$. Recall that the CG estimator is a generalization of the KM estimator. The shaded regions indicate the bounds enclosed by the upper and lower limits (Peterson, 1976).

To estimate the individual survival functions in Fig. 6, we used the TS-RF model combined with the independence copula and the Frank copula with parameters $\theta = -5$ and $\theta = 5$. The shaded regions represent the bounds enclosed by the upper and lower limits given in Inequality 16.

The figures displaying the average survival functions indicate that uncertainty due to the unknown copula increases as time progresses. However, survival probabilities for early times can be estimated with minimal uncertainty, even without prior knowledge of the copula. This observation also holds for individual survival functions. The degree of uncertainty varies by individual, and the right-hand figures demonstrate that for some individuals, survival functions can be well estimated.

## G PROOF OF STRICT PROPERNESS

We prove Theorem 3.

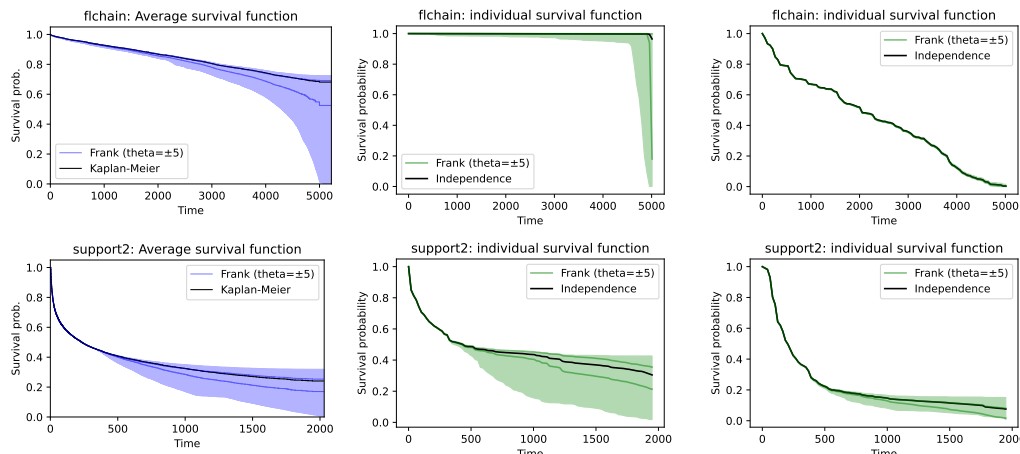

Figure 6: Estimated survival functions with upper and lower bounds for datasets flchain and support2. The left graphs show average survival functions, while the graphs in the center and the right show arbitrary chosen individual survival functions.

*Proof.* We have

$$\mathbb{E}[\mathcal{S}(\{\hat{F}_k(t|x)\}_{k=1}^K, (t,\delta))] - \mathbb{E}[\mathcal{S}(\{F_k(t|x)\}_{k=1}^K, (t,\delta))]$$

$$= \sum_{k=1}^K \int_0^\infty \Pr(T=t, \Delta=k|x)(\log v_k(t|x) - \log \hat{v}_k(t|x))dt$$

$$= \sum_{k=1}^K \int_0^\infty \Pr(\Delta=k|x)\Pr(T=t|\Delta=k|x)(\log v_k(t|x) - \log \hat{v}_k(t|x))dt$$

$$= \sum_{k=1}^K \Pr(\Delta=k|x) \int_0^\infty v_k(t|x)(\log v_k(t|x) - \log \hat{v}_k(t|x))dt$$

$$= \sum_{k=1}^K \Pr(\Delta=k|x) D_{\mathrm{KL}}(v_k(t|x)||\hat{v}_k(t|x))$$

$$\geq 0.$$

Note that Assumption 2 ensures the existence of the KL divergence in the last equality. The last inequality holds if and only if $v_k(t|x) = \hat{v}_k(t|x)$ holds for all $k$ and $t$, which is equivalent to that $F_k(t|x) = \hat{F}_k(t|x)$ holds for all $k$ and $t$. Hence the scoring rule $\mathcal{S}$ is strictly proper. □

## H  ADDITIONAL EXPERIMENTS

The experimental procedures were conducted on a virtual machine possessing a single CPU devoid of any GPU, equipped with a memory of 64 GB, and operating on CentOS Stream 9. The software implementation was achieved using Python 3.11.6 and PyTorch 2.1.2.

**Datasets.**  We used eight datasets, summarized in Table 2, where $N$ denotes the number of data points, and the fourth and fifth columns indicate the numbers of categorical and numerical features in the feature vectors, respectively. The six datasets with $K = 2$ were obtained from the Python package `SurvSet` (Drysdale, 2022). The Framingham (Kannel & McGee, 1979) and PBC (Therneau & Grambsch, 2000) datasets with $K = 3$ were used in (Jeanselme et al., 2023).

In our experiments, all datasets were randomly split into training (65%), validation (15%), and testing (20%) sets. The results reported in this section are the mean and standard deviation over five random splits. We divided the time horizon $[0, t_{\max}]$ into $B - 1$ evenly spaced boundaries

Table 1: Evaluation Metrics

| Problem | Metric Name | Assumption | Metric Type |
|---------|-------------|------------|-------------|
| Density estimation | CJD-Brier | - | Discrimination |
| | CJD-Logarithmic | - | Discrimination |
| | CJD-KS | - | Calibration |
| Survival analysis | Copula-NLL | Copula | Discrimination |
| | Cen-log | Independence | Discrimination |
| | D-calibration | Independence | Calibration |
| | KM-calibration | Independence | Calibration |

Table 2: Real datasets used in our experiments

| Name | $K$ | $N$ | # categorical | # numuerical | censored | max. time |
|------|-----|-----|---------------|--------------|----------|-----------|
| dataDIVAT1 | 2 | 5943 | 3 | 2 | 83.6% | 6225 |
| oldmort | 2 | 6495 | 5 | 2 | 69.7% | 20 |
| Dialysis | 2 | 6805 | 2 | 2 | 76.4% | 44 |
| flchain | 2 | 7874 | 4 | 6 | 72.5% | 5215 |
| support2 | 2 | 9105 | 11 | 24 | 31.9% | 2029 |
| prostateSurvival | 2 | 14294 | 3 | 0 | 94.4% | 119 |
| PBC | 3 | 312 | 5 | 12 | 45.8% | 15 |
| Framingham | 3 | 4434 | 10 | 9 | 56.2% | 8767 |

and added an additional time slot to represent times greater than $t_{\max}$, where $t_{\max}$ is the maximum observed time within the dataset. This setup means that we used $B$ time slots $\{\zeta_b\}_{b=0}^B$ in total. We set $B = 100$ unless otherwise stated.

**Models and Their Hyperparameters.** We used a multi-layer perceptron (MLP) with three hidden layers as a neural network model. The dropout layer was employed with a dropout rate of 0.5, and the ReLU function was utilized as the activation layer. The softmax function served as the output layer. The neural network was trained for 500 epochs using the Adam optimizer (Kingma & Ba, 2015). For each dataset, we performed a hyperparameter search to determine the number of neurons in the hidden layers and the learning rate of the optimizer: the number of neurons was chosen from the set $\{4, 8, 16, 32, 64, 128, 256\}$, and the learning rate was chosen from the set $\{0.00001, 0.0001, 0.001, 0.01, 0.1, 1.0, 10.0\}$.

For tree-based models, a hyperparameter search was also performed: `n_estimators` was chosen from the integers between 100 and 1000, `max_depth` was chosen from the integers between 10 and 50, `min_samples_split` was chosen from the integers between 2 and 64, `min_samples_leaf` was chosen from the integers between 1 and 32, `max_features` was chosen from `sqrt` or `log2`, `criterion` was chosen from `log_loss`, `gini`, or `entropy`, `splitting_rule` was chosen from `CART` or `FourierMMD`, `num_features` was chosen from the integers between 1 and 100, `sample_fraction` was chosen from the numbers between 0.1 and 0.5, `min_node_size` was chosen from the integers between 1 and 10, and `alpha` was chosen from the numbers between 0.01 and 0.3.

**Ablation Study on Hyperparameter $B$.** We conducted an ablation study on the hyperparameter $B$ in our two-step algorithm. This study aimed to evaluate the prediction performance using the TS-DRF model and the Cen-log metric. Specifically, we examined $w_1 \in \{0, 0.5, 1\}$ and $B \in \{20, 40, 60, 80, 100, 120\}$, where $w_1$ is the parameter for the primary event of interest. The graphs in Fig. 7 depict the results.

In Fig. 7, the prediction performances are normalized relative to those with $w_1 = 0.5$ for each $B$. The results indicate that prediction performances varied significantly depending on the choice of the parameter $w_1$ on several datasets when $B$ was small. However, these differences diminished for $B \geq 100$. Therefore, we set $B = 100$ in our experiments.

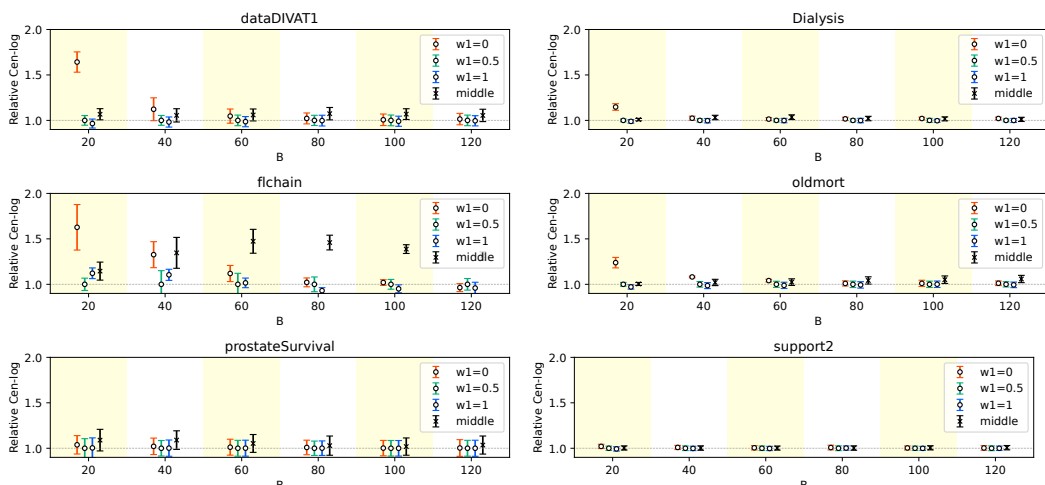

Figure 7: Ablation study on hyperparameter $B$ (lower is better).

The prediction performance labeled 'middle' represents the relative performance of the algorithm presented in (Carrière, 1995) (see Sec. C.3 for details). The results demonstrate that this algorithm performed worse than our algorithm, especially for the datasets dataDIVAT1 and flchain, even when we use a large $B$.

In addition, to complement the results in Sec.7 for the flchain and support2 datasets, we conducted experiments using the dataDIVAT1, Dialysis, oldmort, and prostateSurvival datasets. The outcomes of these experiments are presented in Fig. 8. Consistent with the results from the flchain and support2 datasets, these findings also indicated that no single model consistently outperformed the others across different metrics and datasets.

**Prediction Performance with Frank Copula.**   While the experiments in Sec.7 used the independence copula, we conducted additional experiments using the Frank copula (2) with parameters $\theta = -5$ and $\theta = +5$. Figure 9 presents the results of these experiments. Note that, since the Cen-log, D-calibration, and KM-calibration metrics are valid only if the conditional independence assumption holds (i.e., the independence copula is used), we did not compute these metrics in the experiments. Note also that, since the CJD-Brier, CJD-Logarithmic, and CJD-KS are metrics for density estimation and they measure the prediction performances of the first step of our algorithm, the results were the same as Fig. 5. In other words, the copula $C$ is irrelevant to compute these metrics. Therefore, we included only the results with the Copula-NLL metric in Fig. 9.

Since we used only the Copula-NLL metric in Fig. 9, the Copula-NLL model seemed the best model in this problem setting. However, we should note that the Copula-NLL model did not show the best performance in Fig. 5 with the metrics on the CJD representation (i.e., CJD-Brier, CJD-Logarithmic, and CJD-KS). These results underscore the importance of using multiple evaluation metrics to compare prediction models. Reliance on a single metric could lead to the incorrect conclusion that the Copula-NLL model is the best. A comprehensive evaluation using various metrics provides a more balanced and accurate assessment of model performance.

**Prediction performance on competing risk models.**   We evaluated the prediction performance using the Framingham and PBC datasets with $K = 3$. As baseline methods, we compared our models with those utilized in Jeanselme et al. (2023). Specifically, we compared against: DeepHit (Lee et al., 2018), Deep Survival Machines (DSM) (Nagpal et al., 2021), DeSurv (Danks & Yau, 2022), and Neural Fine-Gray (NeuralFG) (Jeanselme et al., 2023), which is a neural network model extending the Fine-Gray model (Fine & Gray, 1999). For these models, we used the implementations available at https://github.com/Jeanselme/NeuralFineGray/ under MIT license, and performed hyperparameter searches based on the guidelines provided in the source code.

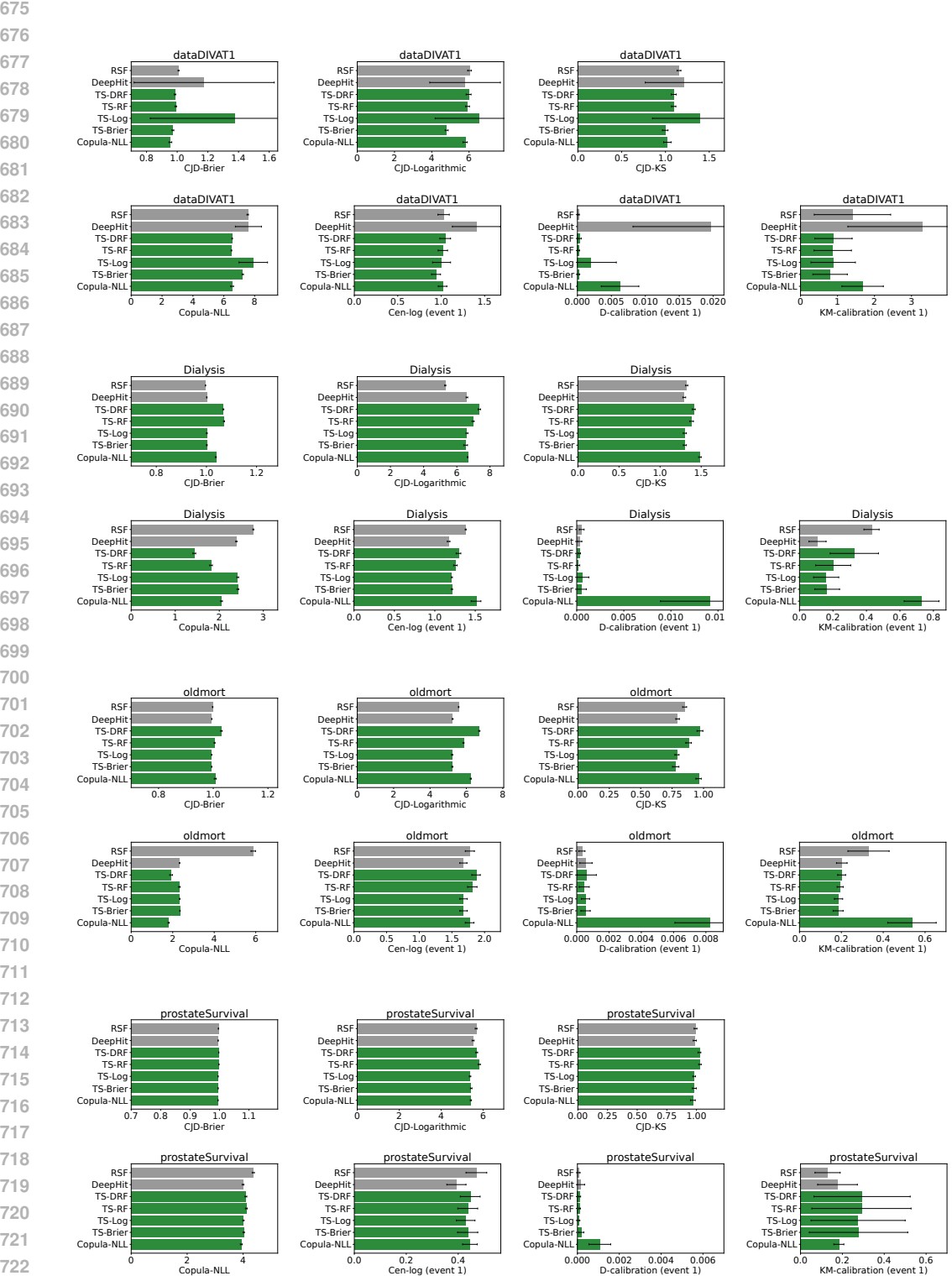

Figure 8: Prediction performance comparison on dataDIVAT1, Dialysis, oldmort, and prostateSurvival datasets with various metrics (lower is better).

We compared our models with DeepHit, DSM, DeSurv, and NeuralFG models using the independence copula, and the results are displayed in Fig. 10. These results demonstrate that our two-step algorithm is competitive with these baseline models.

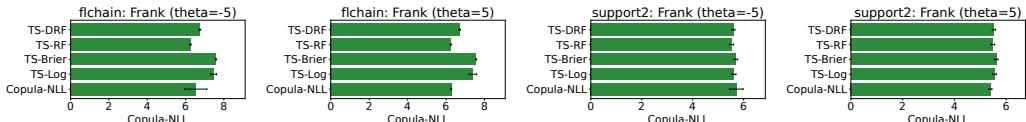

Figure 9: Copula-NLL metric performance on flchain and support2 datasets with Frank copula $\theta = -5$ and $\theta = 5$ (lower is better).

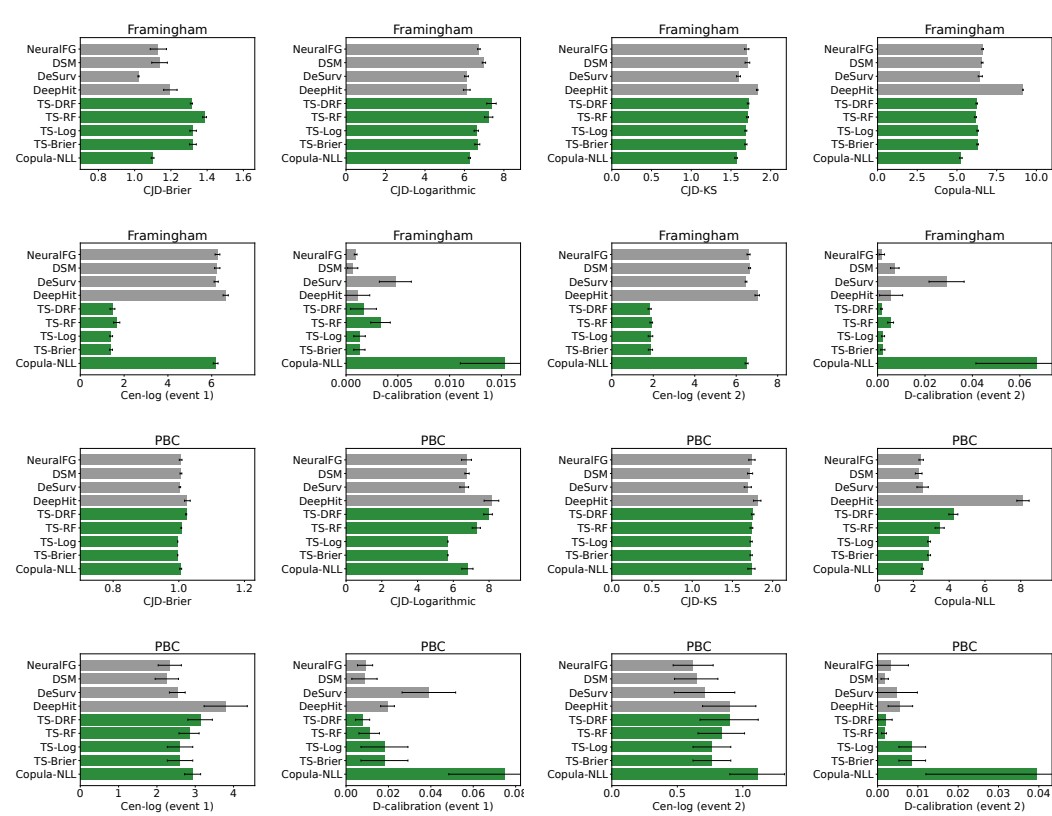

Figure 10: Prediction performance comparison on Framingham and PBC datasets with various metrics (lower is better).

