# OpenReview forum: "Survival Analysis via Density Estimation"
_ICLR.cc/2025/Conference — ICLR 2025 Conference Withdrawn Submission_

### Official Review · Reviewer_7ttV · 2024-11-01

**Soundness:** 3
**Presentation:** 2
**Contribution:** 3
**Rating:** 6
**Confidence:** 4

**Summary:**

In this paper, the  authors  introduce an algorithmm which explain  survival analysis by  density estimation for censored  survival data. The new r approach enhances  the  intrinsic relationship compared with existing methods.  The 2-step  algorithm has  the density estimation result to derive  survival functions. This  paper allows for the application of any density estimation model to  estimate survival functions,  The new algorithm  provides a nice approach for  censored survival data.

**Strengths:**

The paper  is written well. The new framework in the paper  may provide us with  the toolkit  for survival analysis.  The proposed  2-step algorithm can help to  fill out  the gap between density estimation  and survival analysis.  It also offered  several theoretical analyses in detail  and nice properties, and therefore  the authors  established the theoretical results and solid foundations.

**Weaknesses:**

The paper  has good  theoretical results and properties. From  the experiemts of data sets, the paper  shows that no single model consistently outperforms all others across different metrics and datsets.  In addition,  there is no  comparsion of efficiency for the 2-step algorithm and other  algorithms.

**Questions:**

I have several comments and suggestions for the authors to address.

1.  It is worthwhile to add the comparsion of computational costs of differents methods  to check the efficiency.

2.  The paper   is not complete. Please add  the conclusion section.  What is your recommendation of methods for the dataset?

3. From  the experiemts of data sets, the paper  shows that no single model consistently outperforms all others across different metrics and datsets.  Could you elaborate the advantage of the proposed methods compared with existing ones? The authors may discuss specific advantages of the proposed method about flexibility, interpretability  compared to existing methods.

4.  The proposed  framework can be extended to more general case such as interval censoring, double censoing, etc.? The authors  can discuss any challenges they foresee in extending their method to these types of censoring and outline what modifications would be needed.

5.  On p, 1, line 20,  could you explain  "robust approach"?  How to make this conclusion that is robust approach?  It may need  additional evidence to support this claim from the experiemnts.

6.  There are minor typos, etc. in the paper. Please check it carefully. For example, in p. 9, line 464,  what is TS?  Please ensure all acronyms are defined at first use in the paper.

---

### Official Review · Reviewer_8oYh · 2024-11-01

**Soundness:** 4
**Presentation:** 3
**Contribution:** 3
**Rating:** 5
**Confidence:** 4

**Summary:**

This paper introduces a two-step algorithm for survival analysis that reframes the problem as a density estimation task to accommodate censored data. The first step uses a density estimation model to approximate the joint distribution of survival data, and the second step transforms this distribution to yield survival functions. This approach enables the use of various density estimation models to calculate survival functions, broadening the applicability of survival analysis techniques. The authors address key research questions on model flexibility, dependency handling, and uncertainty in survival function bounds, offering a method that can estimate these bounds even when the dependency structure among variables is unknown.

**Strengths:**

- The algorithm can integrate with a wide range of density estimation models, enhancing versatility in survival analysis.
- The method provides upper and lower bounds on survival functions, which is useful when the dependency structure between variables is unknown.

**Weaknesses:**

- The abstract lacks clarity and tends to repeat ideas, especially in the first three sentences, which essentially convey the same concept.
- Why is a discretized time horizon considered in the approach? In practical applications, the time horizon is typically continuous.
- In the second paragraph of the introduction, {$1, 2, \ldots, |\mathcal{T}|$} represents discrete time points, while in the first paragraph of the preliminaries, {$\zeta_0, \zeta_1, \ldots, \zeta_B$} is used for the same purpose. What is the rationale for these notational inconsistencies?
- In Theorem 1, the copula  $C$ is unique if $F_k$ is continuous. However, with a discrete time horizon, the cumulative distribution function $F_k$ is a step function and therefore not continuous.
- Equation references are inconsistent (e.g., "Eq. (4)" versus "equation 7").
- The proposed approach heavily relies on prior knowledge of the copula $C$. More discussion is needed on the robustness of the method with respect to the choice of $C$.

**Questions:**

- In page 3 line 122, $\zeta_0$.
- Could the authors provide additional details on solving the minimization problem discussed on page 6, line 304? They mention that Equation (13) can be solved iteratively for a specific $b$, but it is unclear how this approach accommodates the summation over $b$.
- In Section 4, the authors consider $K=2$ for simplicity. Could they discuss the potential to extend these theoretical results to cases where $K>2$?
- The result in Theorem 2 is dependent on $\tau$. Could the authors elaborate on possible choices for $\tau$ and how these choices may influence the final outcome?

---

### Official Review · Reviewer_ekuo · 2024-11-03

**Soundness:** 3
**Presentation:** 4
**Contribution:** 3
**Rating:** 6
**Confidence:** 3

**Summary:**

The paper demonstrates the connection between density estimation and survival analysis, and introduces how to repurpose existing density estimation strategies to perform survival analysis. It then compares the proposed framework to other survival models.

**Strengths:**

1. The method is well-motivated and well-presented.
2. The proposed method performs competitively.

**Weaknesses:**

1. Besides the models that perform density estimation (DeepHit, random survival forests), the authors could consider adding some other models as their baselines. Some examples include Kaplan-Meier estimator, adaptive Lasso for Cox [1], multi-task learning for survival analysis [2], DeepSurv [3], DRSA [4], Survival-CRPS [5], Inverse-Weighted Survival Games [6].

[1] Zhang, Hao Helen, and Wenbin Lu. "Adaptive Lasso for Cox's proportional hazards model." Biometrika 94.3 (2007): 691-703.
[2] Li, Yan, et al. "A multi-task learning formulation for survival analysis." Proceedings of the 22nd ACM SIGKDD international conference on knowledge discovery and data mining. 2016.
[3] Katzman, Jared L., et al. "DeepSurv: personalized treatment recommender system using a Cox proportional hazards deep neural network." BMC medical research methodology 18 (2018): 1-12.
[4] Ren, Kan, et al. "Deep recurrent survival analysis." Proceedings of the AAAI conference on artificial intelligence. Vol. 33. No. 01. 2019.
[5] Avati, Anand, et al. "Countdown regression: sharp and calibrated survival predictions." Uncertainty in Artificial Intelligence. PMLR, 2020.
[6] Han, Xintian, et al. "Inverse-weighted survival games." Advances in neural information processing systems 34 (2021): 2160-2172.

**Questions:**

None.

---

### Official Review · Reviewer_89bG · 2024-11-03

**Soundness:** 2
**Presentation:** 2
**Contribution:** 2
**Rating:** 3
**Confidence:** 4

**Summary:**

The authors attempt to use density estimation to estimate the conditional probabilities for the competing risks problem. The dependence among the competing survival times is modeled using a copula, but it is debatable whether this copula dependence is appropriate. In survival data, censoring is unavoidable; however, the authors do not mention censoring, leading to the assumption that they treat it as one of the events in the competing risks framework. Both of their claimed research questions Q1 and Q2 focus on K=2, which makes the competing risks trivial since it is necessary to incorporate the censoring class in survival data.

**Strengths:**

Developed the relationship between the density estimation and the marginal probabilities in competing risk problem when the density estimation is available and the dependence is known through copula. Therefore, the marginal probabilities can be obtained on a given sequence.

**Weaknesses:**

The key component for competing risk analysis is to address the dependence among the survival times of different events, and it is directly assumed using copula while it is unjustifiable, especially when K>2.

**Questions:**

1. The paper is about competing risk in survival analysis. The title needs to reflect this.
2. The estimated functions are on the sequence \xi-j, how this sequence is selected should be discussed. In the case of fixed sequence, it is similar to lifetable in survival analysis.
3. The presentation needs improvement. For example, Section 3 discusses  two-step algorithm of the estimation. It is assumed for K competing risk with variables T_1, ..., T_k, but the content in this section is only for K=2; when density estimation is utilized to estimate r_b,k|x, how the censored observations are incorporated.
4. The copula model is employed to address the dependence among the survival times of different events. This dependence is especially not clear when more than two events are considered in the competing risk. It needs to be verified or justified for real applications.
5. The main contribution is soling the equation (13) to obtain the marginal probability estimation F on the given sequence \xi. If the copula C is given and the density is estimated using existing methods, the problem is not  significant.

---

### Official Review · Reviewer_Qsot · 2024-11-07

**Soundness:** 3
**Presentation:** 2
**Contribution:** 1
**Rating:** 3
**Confidence:** 4

**Summary:**

The paper proposed a new approach to performing discrete-time survival analysis under competing risks by reframing it through density estimation. The dependence between the event times was handled by using copula functions. A two-step algorithm was proposed to estimate survival functions. Experiments on various datasets were performed to illustrate the proposed methods.

**Strengths:**

1. The idea of performing survival analysis via density estimation is novel and worthwhile to investigate.
2. The paper is technically solid. I am glad to see the theoretical results.

**Weaknesses:**

1. This paper focused on discrete-time survival under competing risks, which is only a small sub-area of survival analysis. However, I didn't realize it until I read the second paragraph of the Introduction: the title and abstract may exaggerate the contributions. The authors might want to improve the presentation to make things clearer.
2. Estimating the survival functions given the covariates without explicitly modeling their relationship is interesting, but is it really useful? In practice, researchers are often interested in evaluating the effects of covariates on the event times, but the proposed methods cannot do this. I didn't see how the proposed methods could be useful in practice even after I read the experiments section.
3. Related to 2, the experiments seemed to be a weakness. Why didn't you include some traditional models (e.g., the Cox model) as a comparison? If the proposed method cannot beat the Cox model, why should I appreciate it? In addition, the superiority of the proposed method was not clear because it was stated that "no single model consistently outperforms all others across different metrics." Then, why or under what situations should the researchers choose to use your model instead of others?

**Questions:**

See Weaknesses.

---

### Note · Authors · 2024-12-02

**Comment:**

We are grateful to the reviewers for their diligent work and perceptive feedback.

We have decided to withdraw our manuscript from this venue because we found several errors in the Python code used for our experiments, and we need more time to fix them.

**Withdrawal Confirmation:**

I have read and agree with the venue's withdrawal policy on behalf of myself and my co-authors.